

# Quantum state preparation of topological chiral spin liquids via Floquet engineering

**Matthieu Mambrini and Didier Poilblanc⋆**

Laboratoire de Physique Théorique UMR5152, C.N.R.S. and Université de Toulouse,
118 rte de Narbonne, 31062 Toulouse, France

⋆ didier.poilblanc@irsamc.ups-tlse.fr

## Abstract

In condensed matter, Chiral Spin Liquids (CSL) are quantum spin analogs of electronic Fractional Quantum Hall states (in the continuum) or Fractional Chern Insulators (on the lattice). As the latter, CSL are remarkable states of matter, exhibiting topological order and chiral edge modes. Preparing CSL on quantum simulators like cold atom platforms is still an open challenge. Here we propose a simple setup on a finite cluster of spin-1/2 located at the sites of a square lattice. Using a Resonating Valence Bond (RVB) non-chiral spin liquid as initial state on which fast time-modulations of strong nearest-neighbor Heisenberg couplings are applied, following different protocols (out-of-equilibrium quench or semi-adiabatic ramping of the drive), we show the slow emergence of such a CSL phase. An effective Floquet dynamics, obtained from a high-frequency Magnus expansion of the drive Hamiltonian, provides a very accurate and simple framework fully capturing the out-of-equilibrium dynamics. An analysis of the resulting prepared states in term of Projected Entangled Pair states gives further insights on the topological nature of the chiral phase. Finally, we discuss possible applications to quantum computing.


## Contents



# 1 Introduction

The search for spin liquids in condensed matter materials is a very rapidly developing area of two-dimensional (2D) quantum magnetism [1]. Chiral spin liquids (CSLs) are the quantum spin analogs of the topological Fractional Quantum Hall States (FQHS) realized in the strongly interacting two-dimensional (2D) electron gas [2]. A zoo of 2D topological Abelian and non-Abelian CSLs, breaking time-reversal symmetry (T), have already been identified for SU(2) spins on various (frustrated) lattices [3–6], and for spins of higher SU($N$) internal symmetry, $N > 2$ [7,8]. While T-breaking can also occur spontaneously [9–11], most CSLs are stabilized by an explicit T-breaking chiral perturbation.

In 1982, Richard Feynman first proposed that one quantum system could be used to simulate the dynamics of other quantum systems of interest [12]. Nowadays, quantum simulators based on cold atoms on two-dimensional (2D) optical lattices are being realized experimentally with the realistic perspective of emulating simple models of condensed matter physics [13–17] or studying topological quantum matter [18, 19]. Rydberg atoms platforms are also used to realize 2D quantum phases of matter [20] like dimer liquids or spin liquids [21,22]. Recently, a wavefunction with non-Abelian topological order could be prepared using an adaptive circuit on a trapped-ion quantum processor [23].

Designing atomic platforms of quantum spins with a T-breaking term to emulate CSLs is still an open challenge. Very recently, using Floquet engineering, the emulation of the non-Abelian CSL of the Kitaev honeycomb model in the presence of a chiral term has been proposed theoretically using cold atoms or Rydberg simulators [24, 25]. Preparing CSL with full spin-rotational symmetry is also very challenging and of great interest. Proposals based on synthetic gauge fields in N-colors SU($N$) Hubbard models on the square lattice with 1 fermion/site were shown to be effective for $N \geq 3$ [26]. Here we rather focus on the preparation of SU(2)-symmetric spin-1/2 CSLs.

Quench dynamics is an active area of study in recent years [27–32], relevant to a wide range of physical systems, including condensed matter systems, ultracold atomic gases, and quantum field theories. While simple quenches provide a way to probe the non-equilibrium

dynamics of quantum systems, they cannot serve to prepare quantum phases of matter under specific conditions (low temperature, etc...). In contrast, Floquet engineering consisting of applying a fast and strong periodic drive to the system – described by Floquet theory [33] – is an appealing route towards phase/state preparation [34], and even towards non-equilibrium phases without an equivalent counterpart [35]. For example, such a scheme has been proposed to trigger non-trivial energy bands via engineered gauge fields [36]. Our aim is here to design some simple protocols based on Floquet theory to prepare genuine spin-1/2 CSLs. For that purpose, we have developed new theoretical tools to compute faithfully the time evolution of our 2D quantum spin systems in order to address Floquet dynamics in various experimental setups, e.g. a sudden quench of the drive or a semi-adiabatic ramping of the drive. Starting from a paradigmatic spin liquid, the Resonating Valence Bond (RVB) state [37], the finite system is slowly driven out of its equilibrium state, and the time evolution of the system becomes very complex, showing rapid oscillations (micromotion). However, using a high-frequency Magnus expansion [38–41], we analytically construct an effective Floquet Hamiltonian which describes accurately the (stroboscopic) time evolution and makes it much easier to compute numerically.

Under the application of the fast drive, preserving the spin-rotational invariance (singlet character) of the state, we observe a steady increase of the plaquette chirality (more rapid for a quench than for a semi-adiabatic ramping) signaling the emergence of a CSL. Using tensor network techniques in 2D [42], more specifically a chiral Projected Entangled Pair State (PEPS) ansatz [43] (see Appendix A for details), the properties of the final state are uncovered, showing topological features and chiral edge physics characteristic of the Abelian $SU(2)_1$ CSL [44].

## 2 Setups and methods

### 2.1 General considerations on time evolution

The considerations below are fairly general and apply to a closed system (i.e. with no external bath) defined by some initial many-body state $|\Psi_0\rangle$ and its unitary evolution $|\Psi(t)\rangle = U(t, t_0)|\Psi_0\rangle$. In practice, we assume the system is finite. Let us first consider a time-independent Hamiltonian $H$ – typically a Floquet effective Hamiltonian – applied (abruptly) at time $t_0$. The unitary time evolution operator $U(t_0 + t, t_0)$ is just given by $\exp(-iHt)$ (setting $\hbar = 1$ here) and its action on any state $|\Phi\rangle$ can be obtained by constructing the Krylov space $\{H^m|\Phi\rangle\}$ recursively. Then, in principle, one can numerically obtain the exact evolution of the quantum state on a finite system, within machine precision (provided that the computer memory is large enough to accommodate the exponentially large Hilbert space).

In the case of a (more microscopic) time-dependent Hamiltonian $H(t)$, the unitary time evolution operator involves time-ordering, i.e. $U(t_0 + t, t_0) = \mathcal{T}_{t'} \exp\left(-i\int_{t_0}^{t_0+t} H(t')dt'\right)$. In practice, one needs to discretize the time interval by regular time steps $\tau = t/M$. Then, a Trotter-Suzuki (TS) decomposition [45, 46] of the unitary time evolution can be performed in term of elementary gates,

$$U(t_0 + t, t_0) \simeq \exp(-iH(t_M)\tau)\cdots\exp(-iH(t_2)\tau)\exp(-iH(t_1)\tau), \qquad (1)$$

where $t_n = t_0 + (n - \frac{1}{2})\tau$. The time-ordering operator $\mathcal{T}_{t'}$ can then be seen as the $\tau \to 0$ limit of Eq. 1. In practice, $\tau = t/M$ is a small time step, significantly smaller than the smallest relevant time scale in the problem, typically the drive period, ($\tau \ll T_{\text{drive}}$). The TS decomposition involves Trotter errors of order $\tau^2$ which can be controlled by choosing $\tau$ small enough. The states $|\Psi(\tau)\rangle$, $|\Psi(2\tau)\rangle$, ... $|\Psi(M\tau)\rangle$ are obtained recursively using

$|\Psi(n\tau)\rangle = \exp(-iH(t_n)\tau)|\Psi((n-1)\tau)\rangle$. After Taylor expanding $\exp(-iH(t_n)\tau)$ in powers of $H(t_n)$, the calculation is done in the Krylov basis $\{|\Phi_m^n\rangle\}$,

$$|\Psi(n\tau)\rangle = \sum_{m=0}^{m_{max}} \frac{(-i)^m}{m!}|\Phi_m^n\rangle, \tag{2}$$

$$\text{where} \quad |\Phi_m^n\rangle = H(t_n)^m|\Psi((n-1)\tau)\rangle,$$

using the normalisation condition $|\langle\Psi(n\tau)|\Psi(n\tau)\rangle| = 1$ as a test of convergence of the power-expansion up to order $m_{max}$. The Krylov's basis is computed recursively using $|\Phi_m^n\rangle = H(t_n)|\Phi_{(m-1)}^n\rangle$ starting from $|\Phi_0^n\rangle = |\Psi((n-1)\tau)\rangle$, for each $n$.

## 2.2 Initial and targeted spin liquid states

We now move more specifically to our system constituted by an assembly of spin-1/2 on a square lattice. The initial prepared state considered in this work is a simple spin-1/2 nearest-neighbor (NN) Resonating Valence Bond (RVB) state [37]. It can be viewed as an equal-weight superposition of NN singlet (hard-core) covering of the lattice. Conveniently, this spin liquid has a simple Projected Entangled Pair State (PEPS) representation [47, 48] (see Appendix A) which enables to construct it easily on finite systems or on the infinite plane. The NN RVB state has very short-range spin-spin correlations (with a spin correlation length of the order of the lattice spacing) but critical dimer-dimer correlations (on any bipartite lattice). However, any admixture of longer singlets generates a finite dimer correlation length [49, 50]. In that case, the gapped RVB spin liquid acquires $\mathbb{Z}_2$ topological order, similarly to the Kitaev toric code [51].

Most interestingly, possible (closely related) experimental realizations have been proposed recently using Rydberg platforms [22]. Hereafter, we shall consider a $4 \times 4$ finite torus which conveniently offers lattice translation symmetry. However, we believe our setup proposal and our findings are also relevant to lattice systems with open boundaries, a geometry encountered in experiments.

After applying a (fast) drive over a sufficiently long time interval we wish to prepare the system in an (Abelian) Chiral Spin Liquid (CSL) phase, possibly at low effective temperature (if not in the ground state). It is known that such a phase is realized in a simple chiral spin-1/2 Heisenberg Hamiltonian on the square lattice, which we view as a "target" Hamiltonian,

$$H_{target} = H_0 + \lambda_{chiral} i \sum_{\square} (P_{ijkl} - P_{ijkl}^{-1}), \tag{3}$$

where the last term involves cyclic (let's say anti-clockwise) permutations $P_{ijkl}$ on all plaquettes $\square$. $P_{ijkl}$ performs a shift along the $ijkl$ ring of any spin state $|\alpha\rangle_i \otimes |\beta\rangle_j \otimes |\gamma\rangle_k \otimes |\delta\rangle_l$ mapping it to $|\beta\rangle_i \otimes |\gamma\rangle_j \otimes |\delta\rangle_k \otimes |\alpha\rangle_l$. This term breaks time-reversal (T) and parity (P). Note that the cyclic permutation on the (oriented) plaquette $(i, j, k, l) \equiv (i, i+e_x, i+e_x+e_y, i+e_y)$ has a simple expression in terms of the spin degrees of freedom,

$$i \sum_{\square} (P_{ijkl} - P_{ijkl}^{-1}) = 2 \sum_{\square} \Big\{ \mathbf{S}_i \cdot (\mathbf{S}_{i+e_x} \times \mathbf{S}_{i+e_y}) + \mathbf{S}_{i+e_x} \cdot (\mathbf{S}_{i+e_x+e_y} \times \mathbf{S}_i)$$
$$+ \mathbf{S}_{i+e_x+e_y} \cdot (\mathbf{S}_{i+e_y} \times \mathbf{S}_{i+e_x}) + \mathbf{S}_{i+e_y} \cdot (\mathbf{S}_i \times \mathbf{S}_{i+e_x+e_y}) \Big\},$$

as can be verified by a direct comparison of the $2^4 \times 2^4$ matrices in the $S_z$ basis on both sides of the above identity.

The other term $H_0$ is the spin-1/2 quantum antiferromagnet with NN antiferromagnetic coupling $J_1$ and, optionally, next-NN (NNN) antiferromagnetic coupling $J_2 < J_1$,

$$H_0 = J_1 \sum_{(ij)} \mathbf{S}_i \cdot \mathbf{S}_j + J_2 \sum_{((ij))} \mathbf{S}_i \cdot \mathbf{S}_j. \tag{4}$$

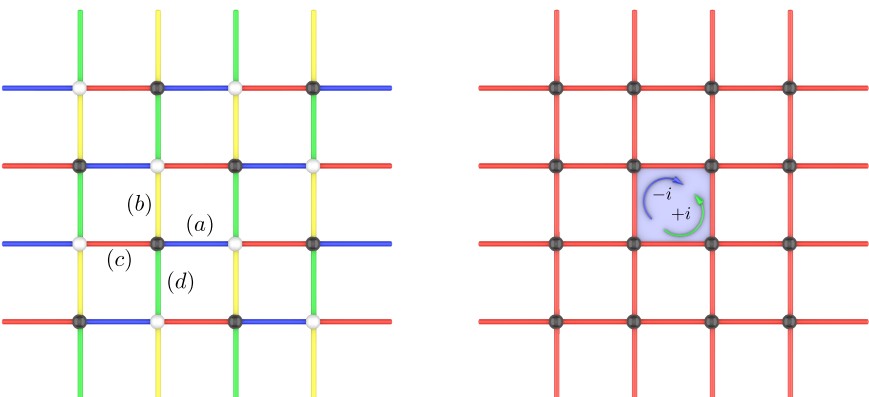

Figure 1: *Left panel* – Sketch of the $H_{\text{drive}}$ Hamiltonian given by Eq. 5. Dephased terms $H_a, H_b, H_c$ and $H_d$ are represented as blue, yellow, red and green bonds. Note that the bond modulations around the sites A (black dots) and sites B (white dots) of the (bipartite) square lattice are dephased by $\pi$. *Right panel* – Floquet effective Hamiltonian $H_F$ given by Eq. 7 acting on square plaquettes as circular anticlockwise (resp. clockwise) permutations with a factor $i$ (resp. $-i$).

In fact, the phase diagram of this model hosts the desired (gapped) topological CSL phase over a significant region of its parameter space [52–55], even for a vanishing frustrating coupling $J_2 = 0$, but for finite $\lambda_{\text{chiral}}$. Interestingly, the corresponding CSL phase can be represented accurately by PEPS [43, 44, 53, 56], which conveniently give crucial insights on the topological features and on the nature of the chiral edge modes, via the bulk-edge correspondence PEPS theorem [57].

## 2.3 Fast chiral periodic drive

The (ideal) monochromatic drive we consider involves dephased sinusoidal modulations of the spin-spin interaction on the NN $a$, $b$, $c$ and $d$ bonds around all the sites of one of the two sublattices (let's say $A$). More precisely, we assume,

$$H_{\text{drive}}(t) = H_a(t) + H_b(t) + H_c(t) + H_d(t), \tag{5}$$

$$H_a(t) = J \cos(\omega t) \sum_{i \in A} \mathbf{S}_i \cdot \mathbf{S}_{i+e_x},$$

$$H_b(t) = J \cos(\omega t - \pi/2) \sum_{i \in A} \mathbf{S}_i \cdot \mathbf{S}_{i+e_y},$$

$$H_c(t) = J \cos(\omega t - \pi) \sum_{i \in A} \mathbf{S}_i \cdot \mathbf{S}_{i-e_x},$$

$$H_d(t) = J \cos(\omega t - 3\pi/2) \sum_{i \in A} \mathbf{S}_i \cdot \mathbf{S}_{i-e_y},$$

where $J$ is the coupling constant and $e_x$, $e_y$ are the unit vectors along the crystal axis. As shown in Fig. 1 the bond pattern is staggered leading to two types of plaquettes $adcb$ and $cbad$. Interestingly, $H_{\text{drive}}(t)$ is chiral, breaking parity $P$ and time-reversal $T$ but not the product PT. Indeed, $P$ (reflection w.r.t. a crystal axis) changes the plaquette bond ordering, e.g. $adcb \rightarrow abcd$, i.e. the sign of the phases in Eq. 5. Time-reversal $T : t \rightarrow -t$ has the same effect. Note that the sign of the chirality in the two types of plaquettes is the same so that the overall chirality is uniform. Note also that the initial state is a spin-singlet and the Hamiltonian in Eq. 5 is invariant under SU(2) spin rotations. The time evolution therefore occurs in the spin-singlet manifold.

It can be noticed that the sequence we are considering has similarities to the driving protocol introduced in Ref. [58] for a tight-binding model of non-interacting particles on the square lattice, in which hopping amplitudes are varied in a spatially homogeneous but chiral, time-periodic way resulting in chiral edge states.

Here the drive is assumed to be very fast compared to the natural time scale $\hbar/J$, i.e. $\hbar\omega \gg J$. Hereafter we set $\hbar = 1$ (unless useful to keep it) and take $\omega = 2\pi f$ with a frequency $f = 10$. Hence, the drive time-periodicity $T_{\text{drive}}$ is 0.1 (in units of $1/J$ taken as the unit of time throughout), which is the smallest time scale in the problem. Therefore, to compute the time evolution using a TS decomposition we have to divide the drive period into a sufficiently large number $N_\tau$ of time steps multiple of $\tau = T_{\text{drive}}/N_\tau$.

Such a dephased monochromatic drive may be implemented on experimental cold atom platforms [19]. We shall discuss variations of this setup based on square modulations, also adapted to experiment, later in the conclusion section. Note that the spin-spin interaction may be challenging to realize on analog platforms. Digital (noisy) quantum computers may be an alternative.

## 2.4 Effective Floquet Hamiltonian

In the case of fast drives, different types of high-frequency expansions can be performed in powers of $1/\omega$. For practical reasons (which will become clear soon), we follow here the Magnus expansion for the stroboscopic Floquet Hamiltonian [41]. In that case, the effective (time-independent) Floquet Hamiltonian describes the slow motion at stroboscopic times, $T_p = p T_{\text{drive}}$, $p \in \mathbb{N}$. The fast micromotion between stroboscopic times is set by the periodic kick operator (not computed) which vanishes at $t = T_p$ for all integer $p$. The drive Hamiltonian has no constant part, $(1/T_{\text{drive}}) \int_0^{T_{\text{drive}}} H_{\text{drive}}(t) dt = 0$, and just single-harmonics components,

$$H_{\text{drive}}(t) = \cos(\omega t)H_x + \sin(\omega t)H_y \,, \tag{6}$$

$$\text{where} \quad H_x = J \sum_{i \in A}(\mathbf{S}_i \cdot \mathbf{S}_{i+e_x} - \mathbf{S}_i \cdot \mathbf{S}_{i-e_x}) \,,$$

$$\text{and} \quad H_y = J \sum_{i \in A}(\mathbf{S}_i \cdot \mathbf{S}_{i+e_y} - \mathbf{S}_i \cdot \mathbf{S}_{i-e_y}) \,.$$

The leading-order effective Floquet Hamiltonian is therefore [41],

$$H_F^0 = \frac{i}{2\omega}[H_x, H_y]$$

$$= J_F \, i \sum_{\square}(P_{ijkl} - P_{ijkl}^{-1}) \,, \tag{7}$$

where $J_F = \frac{J^2}{4\omega}$ and $P_{ijkl}$ is the 4-site cyclic permutation which applies to all plaquettes $\square$. The details of the derivation is left in the Appendix B. It is interesting that the next order of the expansion is $\mathcal{O}(1/\omega^3)$ so that we expect $H_F^0$ to capture accurately the stroboscopic motion.[1] $H_F^0$ is a chiral Hamiltonian, acquiring its P and T symmetries from the microscopic model $H_{\text{drive}}$: it transforms into its opposite under $P$ or $T$ and is invariant under $PT$. Interestingly, it is fully symmetric – invariant under lattice translations and under all discrete symmetries of the $C_{4v}$ point group – in contrast to $H_{\text{drive}}$ which bears two types of plaquettes (of same chirality) as seen Fig. 1. The Floquet Hamiltonian is introducing the longest time scale in the problem $t_F = \hbar/J_F = (4\hbar\omega/J)\frac{\hbar}{J}$. Hence we are in the regime where,

$$t_F \gg \hbar/J \gg T_{\text{drive}} \,, \tag{8}$$

---

[1]Eq. (42) of Ref. [41] suggests that, in the absence of a time independent part i.e. when $H_0 = 0$, the expansion involves only nested commutators containing the same number of $H_1$ and $H_{-1}$ Fourier components of the Hamiltonian, i.e. odd powers of $1/\omega$.

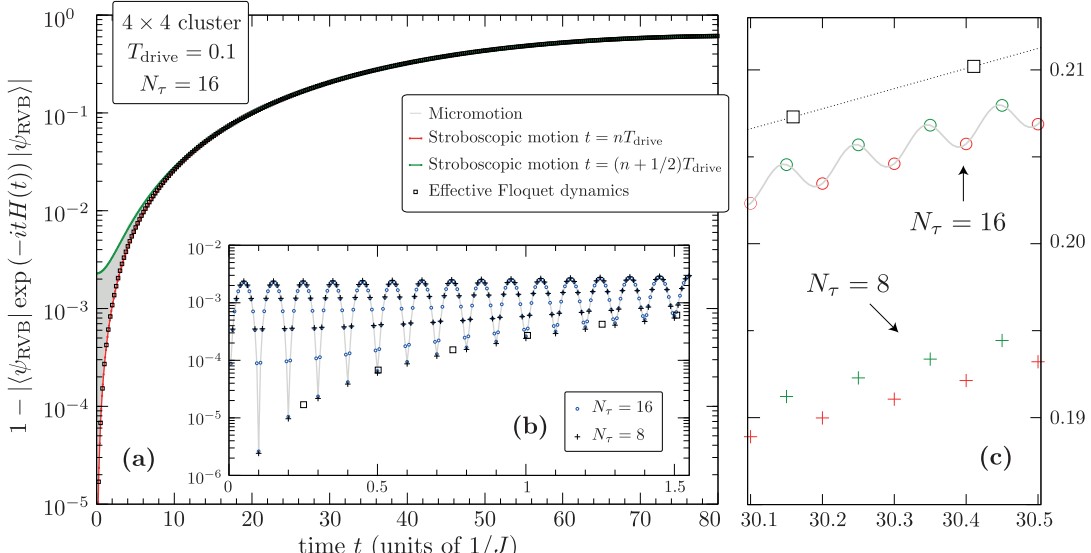

Figure 2: Infidelity $1-|\langle\Psi_{\mathrm{RVB}}|U(t,0)|\Psi_{\mathrm{RVB}}\rangle|$ w.r.t. the initial state, the NN RVB spin liquid, computed on a $4\times4$ torus in the case of a pure chiral drive $H(t)=H_{\mathrm{drive}}(t)$. (a) Comparison of the unitary evolutions with the drive Hamiltonian and with the (time-independent) effective Floquet Hamitonian obtained on a $4\times4$ cluster. Zooms at small times (b) or around $t=30$ (c) are shown. Trotter errors are negligible at short times (b) but become sizeable for $N_\tau=8$ at intermediate times (c).

with 3 well-separated time (or energy) scales. Next, we shall also consider the case where a constant term $\kappa H_0$ is added to the drive Hamiltonian leading to a more generic Floquet Hamiltonian $H_F$. For convenience we shall denote $|\Psi_{\mathrm{Floquet}}(t)\rangle$ the state evolved according to the Floquet Hamiltonian $\exp(-iH_F t)|\Psi_{\mathrm{RVB}}\rangle$, to be distinguished from the state $|\Psi_{\mathrm{drive}}(t)\rangle=\mathcal{T}_{t'}\exp\left(-i\int_0^t(\kappa H_0+H_{\mathrm{drive}}(t'))dt'\right)|\Psi_{\mathrm{RVB}}\rangle$ following the exact evolution.

## 2.5 Quench and adiabatic evolution

The objective of the setup is to emulate the low-energy physics (possibly the groundstate) of the (static) chiral antiferromagnetic Hamiltonian $H_{\mathrm{target}}$ on the square lattice defined in Eq. 3. It is known that the phase diagram of this model bears a wide CSL phase, even for a vanishing frustrating coupling $J_2=0$. While emulating the NN coupling $J_1$ between the spins can be done experimentally, generating the necessary chiral term is a difficult task. For this aim, we add the Hamiltonian $H_0$ to the drive Hamiltonian with some (small) amplitude $\kappa$ to be adjusted,

$$H(t)=\kappa H_0+\lambda(t)H_{\mathrm{drive}}(t).\tag{9}$$

We also consider the possibility of slowly ramping the amplitude $J$ of the drive by some time-dependent factor $0\le\lambda(t)\le1$. Two scenarios will be considered, either (i) a quench setup where $\lambda(t)=1$ from the very beginning of the drive application at $t=0$ or (ii) a quasi-adiabatic setup where $\lambda$ is increased smoothly from 0 at $t=0$ to 1 at $t=t_{\mathrm{ramp}}$.

The most interesting regime to consider is the so-called strong (and fast) drive regime $\kappa J_1\ll J$, i.e. $\kappa\ll1$ if $J$ and $J_1$ are both taken as energy reference, $J=J_1=1$. If $\lambda(t)$ is not too small (i.e. excluding the short initial branching of the drive), the new high-frequency effective Floquet Hamiltonian is simply,

$$H_F(t)\simeq\kappa H_0+\lambda^2(t)J_F\,i\sum_\square(P_{ijkl}-P_{ijkl}^{-1}),\tag{10}$$

which is exactly of the form of the target Hamiltonian (3). Note that in the case of a slow ramp, the Floquet Hamiltonian itself acquires a slow time dependence. When a quench is realized, $\lambda(t) = 1$, one can choose $\kappa$ so that $H_F = \kappa H_{\text{target}}$. Using (3) and (10), one gets

$$\kappa = \frac{J_F}{\lambda_{\text{chiral}}} = \frac{J^2}{4\omega\lambda_{\text{chiral}}} \ll 1 \,. \tag{11}$$

Note that, because of the presence of $H_0$ in Eq. 9, there are corrections to Eq. 10 of order $\mathcal{O}(1/\omega)$ instead of $\mathcal{O}(1/\omega^3)$ (see Eq. (42) of Ref. [41] and Appendix B) . However, these additional terms have amplitudes $\kappa J_1 J/\omega$ i.e, from Eq. 11, of the order of $J_1 J^2/\omega^2$ and, hence, can be neglected compared to the terms in Eq. 10.

## 3   Results

### 3.1   Validity of the Magnus high-frequency expansion

In order to check the relevance of the Floquet Hamiltonian to describe the effective dynamics, we first consider the simplest possible setup, namely the application of the fast chiral drive alone on the NN RVB state, i.e. we set $x = 0$ and $\lambda(t) = 1$ in Eq. 9. Our initial state is invariant under $P$ and $T$ while $H_F$ is odd so we expect $\langle H_F \rangle \equiv \langle \Psi_{\text{Floquet}}(t)|H_F|\Psi_{\text{Floquet}}(t)\rangle = 0$, i.e. the stroboscopic motion occurs in the zero-chirality manifold. Considering a $4 \times 4$ finite cluster with PBC (torus), we have checked numerically that this property is even exactly realized during the micromotion under $H_{\text{drive}}(t)$. Indeed, if one considers any 4-site oriented plaquette $(i, j, k, l)$, $\langle \Psi(t)|\mathbf{S}_i \cdot (\mathbf{S}_j \times \mathbf{S}_k)|\Psi(t)\rangle = -\langle \Psi(t)|\mathbf{S}_k \cdot (\mathbf{S}_l \times \mathbf{S}_i)|\Psi(t)\rangle$ at all times, i.e facing triangles in all plaquettes have always opposite chiralities (of small magnitudes). Hence, the overall plaquette chirality $\langle \Psi(t)|i(P_{ijkl} - P_{ijkl}^{-1})|\Psi(t)\rangle$ is exactly vanishing at all times.

To have a more precise comparison between the exact and the effective dynamics, we have computed the infidelity w.r.t. to the initial state, i.e. $\mathcal{I}(t) = 1 - |\langle \Psi(t)|\Psi_{\text{RVB}}\rangle|$, where $|\Psi(t)\rangle$ is computed using either $H_{\text{drive}}(t)$ ($|\Psi(t)\rangle = |\Psi_{\text{drive}}(t)\rangle$) or $H_F$ ($|\Psi(t)\rangle = |\Psi_{\text{Floquet}}(t)\rangle$), starting from the same initial state $|\Psi_{\text{RVB}}\rangle$ at $t_0 = 0$. The results in Fig. 2(a) show an excellent agreement. A zoom at small time in Fig. 2(b) shows a clear oscillation within every time period (micromotion), while the motion at (discrete) stroboscopic times $(p/N_\tau + n)T_{\text{drive}}$, $p$ fixed, falls on smooth curves when varying $n$. The result at $p = 0$, that is at times exact multiples of $T_{\text{drive}}$, compares very well with the calculation obtained using $H_F$, i.e. approximating $|\Psi(t)\rangle$ by $|\Psi_{\text{Floquet}}(t)\rangle = \exp(-iH_F t)|\Psi_{\text{RVB}}\rangle$.

However, a zoom at a longer time $t \sim 30$ in Fig. 2(c) shows small deviations between the stroboscopic motion computed with the drive Hamiltonian using two choices of $N_\tau$ and the effective Floquet motion (computed exactly). These deviations may have two origins: first, the Trotter errors in the time discretization of the drive oscillations and, secondly, the neglected higher order terms in the Floquet Hamiltonian. The truncation of the Floquet Hamiltonian leads to *relative* errors of order $1/\omega^2$, i.e. $\sim 2 \times 10^{-4}$ which can be neglected. In contrast, the Trotter errors are still sizeable at $N_\tau = 8$. Using $N_\tau = 16$ Trotter steps per drive period seems to give already a much better accuracy, typically within 2% of the (exact) effective Floquet motion. Interestingly, we also observe that the amplitude of the micromotion oscillations becomes quite small compared to the average behavior, except at very small time.

### 3.2   Quench of the fast drive

In this section we shall assume the drive Hamiltonian (10) is suddenly switched on at $t = 0$ on the RVB initial state, i.e. we assume $\lambda(t) = 1$, independent of $t$. In that case, the effective

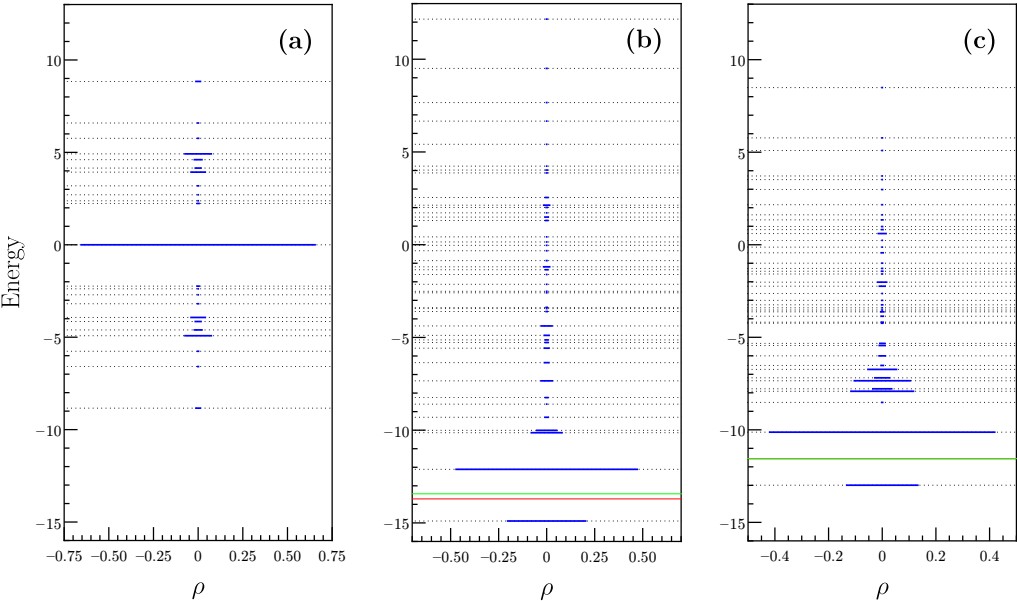

Figure 3: Diagonal ensembles: comparison between distributions of diagonal density matrices $\rho_\mu$ vs $E_\mu$, the eigenvalue spectrum of $H_{\text{target}} = \frac{\lambda_{\text{chiral}}}{J_F} H_F$; (a) pure drive $\kappa = 0$ i.e. $H_{\text{target}}$ given by $J_1 = J_2 = 0$ and $\lambda_{\text{chiral}} = 0.5$; (b) $\kappa \neq 0$ given by Eq. 11, $H_{\text{target}}$ given by $J_1 = 1$, $J_2 = 0$ and $\lambda_{\text{chiral}} = 0.7$; (c) Same as (b) but for $J_2 = 0.4$ and $\lambda_{\text{chiral}} = 0.5$. Energy levels are represented by black dotted lines and their weight by segments whose 1/2-length is equal to $\rho$. Note that $\sum_\mu \rho_\mu = 1$. The first singlet excited state, forming together with the GS the pair of relevant CFT states [52], is shown by a green line in (b) and (c). In both cases, note the presence of a low energy singlet state with $(\pi, \pi)$ momentum (shown in red). In case (c), green and red states are very close but non-degenerate.

motion follows the unitary evolution given by $U(t) = \exp(-i\kappa H_{\text{target}} t)$, which depends on the control parameters $J_1$, $J_2$ and $\lambda_{\text{chiral}}$. The case $J_1 = J_2 = 0$ corresponds to the previously studied case $H_0 = 0$ i.e. a purely oscillatory drive. If the constant term $H_0$ is present we set $J_1 = 1$.

### 3.2.1 Long-time average and diagonal ensembles

Since the effective Floquet dynamics is expected to be very accurate, one can use it to investigate the physics in the long time limit, after a quench of the drive Hamiltonian, i.e. assuming $\lambda = 1$ in Eq. 10. Strictly speaking, the approximate effective Hamiltonian computed from the Magnus expansion does not capture heating effects. The system is expected to heat up to an infinite temperature state in the $t \to \infty$ limit for any finite $\kappa > 0$, see e.g., Ref. [35]. Note however that on a finite system of $N$ sites, the bandwidth of the many-body energy spectrum scales like $W_\omega \sim aNJ_F = (a/4)NJ^2/\omega$, where $a$ of order 1. Hence, at large enough frequency, the many-body spectrum does not reach the boundary of the Brillouin-Floquet energy zone of extension $\omega$. In our case $a \sim 2.4$ (see numerical computations below) and $W_\omega \simeq 0.15J$, significantly smaller than $\omega = 20\pi J$. We therefore expect an extremely long prethermalization regime [35].

Decomposing the initial (non-chiral) spin liquid state in terms of the Floquet eigenstates $|e_\mu\rangle$ of $H_F$, $|\Psi_{\text{RVB}}\rangle = \sum_\mu w_\mu |e_\mu\rangle$, the infinite-time average of observables $\mathcal{O}$ can be written in

terms of the diagonal density matrix $\rho_{\text{diag}} = \sum_\mu \rho_\mu |e_\mu\rangle\langle e_\mu|$ [41],

$$\langle \mathcal{O} \rangle_{\text{ave}} = \sum_\mu \overline{\mathcal{O}}_{\mu\mu} \rho_\mu \,, \tag{12}$$

where $\rho_\mu = |w_\mu|^2$. Within a time period, the wavefunction $|\Psi(nT + t)\rangle$ is subject to a unitary rotation $\exp(-iK(t))$ where $K(t)$ is a periodic kick operator (see explicit expression in Appendix C). Hence, the infinite-time average in Eq. 12 involves the averaged operator,

$$\overline{\mathcal{O}} = \frac{1}{T} \int_0^T \exp(iK(t)) \mathcal{O} \exp(-iK(t)) dt \,. \tag{13}$$

Note that, for the observables used in this work, the effect of the kick rotations can be neglected so that $\overline{\mathcal{O}}_{\mu\mu} \simeq \mathcal{O}_{\mu\mu}$ in Eq. 12. Note also that, since the RVB state is a singlet state invariant under all space group operations, its decomposition in terms of Floquet eigenstates is limited to the most symmetric singlet sector.

When $\kappa = 0$ (sinusoidal drive alone) the initial RVB state is a highly excited state of $H_F^0$. Indeed, $H_F^0$ has a symmetric spectrum with $\pm E_\mu$ opposite eigenvalues associated to pairs of complex-conjugate eigenvectors $|e_\mu^0, -\rangle = |e_\mu^0, +\rangle^*$. The RVB state being a real (T-invariant) wavefunction, its decomposition is of the form

$$|\Psi_{\text{RVB}}\rangle = \sum_\mu c_\mu^0 \left( |e_\mu^0, -\rangle + |e_\mu^0, +\rangle \right), \tag{14}$$

where $c_\mu^0$ are real coefficients. The $E_\mu \leftrightarrow -E_\mu$ symmetry of the weight distribution is indeed clear in Fig. 3(a). One also immediately gets $\langle H_F^0 \rangle_{\text{ave}} = 0$, in agreement with the fact that the chirality remains zero under time evolution, as noted before. Interestingly, we see that the initial RVB state has a very large overlap with the zero energy states located at the middle of the many-body spectrum of $H_F^0$. For larger $L \times L$ systems of increasing $N = L^2$ sites, a broader weight distribution is expected, still centered around the spectrum center $E = 0$, so that the system can be viewed as being at infinite temperature. A more exotic scenario would be realized if the RVB state keeps a finite overlap on the $E = 0$ state manifold suggesting the existence of low-entangled "scar" states [59,60] (i.e. following the area law) in the center of the spectrum and, hence, a lack of thermalization.

Weight distributions for $\kappa \neq 0$, i.e. when a constant term is present in the drive Hamiltonian, are shown in Fig. 3(b,c), in the cases corresponding to $H_{\text{target}}$ given by $J_1 = 1$, $J_2 = 0$ and $\lambda_{\text{chiral}} = 0.7$, and by $J_1 = 1$, $J_2 = 0.4$ and $\lambda_{\text{chiral}} = 0.5$, respectively. Then, the spectrum of $H_{\text{target}} = \frac{1}{\kappa} H_F$ is no longer symmetric and one gets finite values for the time-averaged plaquette chirality $\text{Im}\langle P_{ijkl} \rangle_{\text{ave}} \sim 0.2496$ and $0.2334$ in the (b) and (c) cases, respectively. The GS and the 1st singlet excited state (shown as a red line in Figs. 3(b,c)) have been identified from a Conformal Field Theory (CFT) construction as the two relevant states forming the CSL doubly-degenerate GS manifold on the infinite-size torus [52]. Note that the RVB state has no overlap with the second CFT state which possesses different quantum numbers. However, large weights are found on the GS and on the next lowest *fully symmetric* singlet eigenstates of $H_F$.

### 3.2.2 Time evolution at small and intermediate times

We now turn to the actual computation of the time evolution in the case of a quench of the drive. Results obtained on the $4 \times 4$ torus are shown in Fig. 4(a). The constant $H_0$ of the quenched Hamiltonian (10) involves either $J_1 = 1$ and $J_2 = 0$ or $J_1 = 1$ and $J_2 = 0.4$. The amplitude $\kappa$ of $H_0$ is set using Eq. 11 i.e. $\kappa = J^2/(4\omega\lambda_{\text{chiral}})$, and we choose either $\lambda_{\text{chiral}} = 0.7$

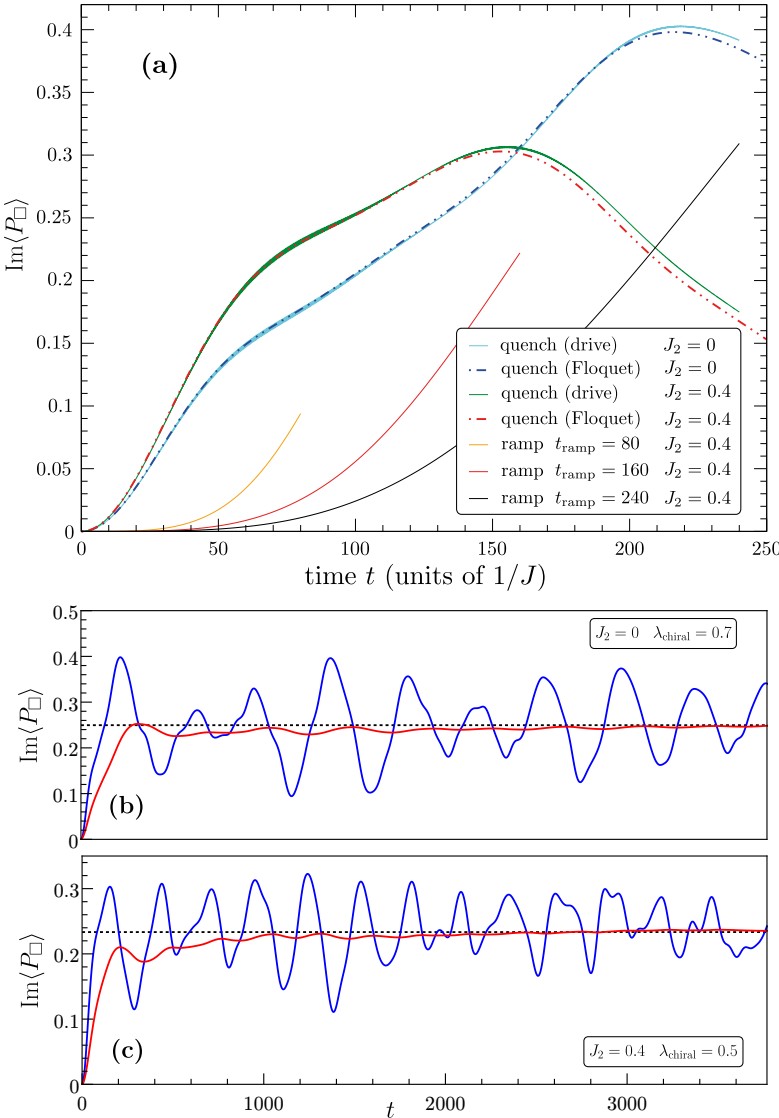

Figure 4: Plaquette chirality vs time for a quench of the drive computed on a $4 \times 4$ torus. Parameters are $J_1 = 1$, $J_2 = 0$ and $\lambda_{\text{chiral}} = 0.7$ or $J_1 = 1$, $J_2 = 0.4$, and $\lambda_{\text{chiral}} = 0.5$. (a) Short and intermediate times behavior; all computations involving the fast drive are done with $N_\tau = 16$ intervals in each drive period but only data at times $kT_{\text{drive}}$ and $(k + 1/2)T_{\text{drive}}$, $k \in \mathbb{N}$, are reported since the amplitude of the oscillations within each time period (micromotion) remains very small. Computations using the effective Floquet dynamics with $U(t) = \exp(-i\kappa H_{\text{target}} t)$ shown as dashed-dotted curves agree very well. Results involving a ramping of the drive are also shown for comparison. (b,c) Long time behavior obtained using the effective Floquet dynamics; the chirality (blue curve) is shown to oscillate around the average value (black dashed line) computed independently using Eq. (12). The time average $\frac{1}{t} \int_0^t \text{Im}\langle P_\square \rangle(\tau) \, d\tau$ is displayed as a red line.

when $J_2 = 0$ or $\lambda_{\text{chiral}} = 0.5$ when $J_2 = 0.4$, corresponding to the two realistic $H_{\text{target}}$ whose spectra are shown in Fig. 3(b,c). We observe a rapid increase of the plaquette chirality, initially vanishing in the RVB state. Note that the amplitude of the micromotion fast oscillations remains quite small, barely visible on the scale of the plot, so that the full time evolution can be compared directly to the effective Floquet dynamics, not only at stroboscopic times. We observe that the effective Floquet approach is able to capture accurately the slow variations of the chirality expectation value up to long times. Even more, we believe that the small deviation at intermediate times with the full time-dependent simulations is primarily due to the Trotter error in the later rather than to the truncation of the Floquet Hamiltonian at order $1/\omega$. At times $t > t_F$ we observe in Fig. 4(b) some strong oscillations around the expected long-time average values computed from Eq. 12. This is clearly a finite size effect and one expects the amplitude of the oscillations to decrease rapidly with increasing system size. In any case, we believe that quenching the drive Hamiltonian is not the correct procedure to achieve the goal of preparing the system in the final CSL state and we now move to a different setup.

### 3.3 Adiabatic ramping of the fast drive

The new setup we shall follow now consists in branching the fast drive in a quasi-adiabatic way. For that we have chosen a simple linear ramp,

$$\lambda(t) = t/t_{\text{ramp}}, \tag{15}$$

for $t \in [0, t_{\text{ramp}}]$. In a truly infinitely-slow ramping $\lambda(t)$ from 0 to 1 (e.g. taking $t_{\text{ramp}} \to \infty$), the state $|\Psi(t)\rangle$ is expected to follow adiabatically the GS of

$$H_F(t) \simeq \kappa[H_0 + \lambda^2(t)(H_{\text{target}} - H_0)], \tag{16}$$

provided that $|\Psi(0)\rangle$ is the GS of $H_0$. In practice, deviations from this ideal scenario are present. First, the parent Hamiltonian of the RVB state in the thermodynamic limit involves longer range interactions [47, 48] than the NN and NNN interactions considered here. However, we have checked that, on a finite system, the RVB state is a good approximation of the exact GS of $H_0$ for $J_2 = 0.4J_1$, with an overlap of order 0.9960 on a $4 \times 4$ cluster. Secondly, $t_{\text{ramp}}$ is finite leading to a deviation from the adiabatic behavior. It is then interesting to try to quantify the effect of a finite ramp time. Note that, although the branching of the drive is linear in $t$, the effective coupling to $H_{\text{target}}$ in (16) is smoother, increasing only as $t^2$ at small time. Note also that the initial RVB state has a sizeable overlap with the targeted final state, as shown in Fig. 3(c), which should enable an adiabatic evolution in a shorter time.

We have computed the time evolution (using $N_\tau = 16$ TS steps per drive period $T_{\text{drive}} = 0.1$) and the emergence of the plaquette chirality for three ramp times of increasing length, $t_{\text{ramp}} = 80, 160,$ and 240 as plotted in Fig. 4(a), showing a smooth behavior of the observable, in contrast to the quench setups. As expected, the increase of the chirality is slower and slower when increasing the ramp time but the final chirality reached at $t = t_{\text{ramp}}$ becomes larger and larger.

Fig. 5(a) shows similar data as a function of the reduced time $t/t_{\text{ramp}}$. The adiabatic groundstate $|\Psi_{\text{adiab}}(\lambda)\rangle$ is obtained by following the GS of the Floquet Hamiltonian (16), varying $\lambda$ "by hand" from 0 to 1. It should be reached asymptotically by taking the limit $t_{\text{ramp}} \to \infty$ keeping the ratio $t/t_{\text{ramp}} = \lambda$ fixed. In fact, the GS chirality of $H_{\text{target}}$ is $\sim 0.478$ while the achieved chirality at $t = t_{\text{ramp}} = 320$ is only $\sim 0.383$. This indicates that longer ramp times are needed to achieve the goal of approaching more closely the GS of $H_{\text{target}}$. However, running the full drive simulation over longer times is expensive in CPU-time and, in fact, not necessary. Indeed, we have checked that for $t_{\text{ramp}} = 320$, the effective evolution with the slowly varying

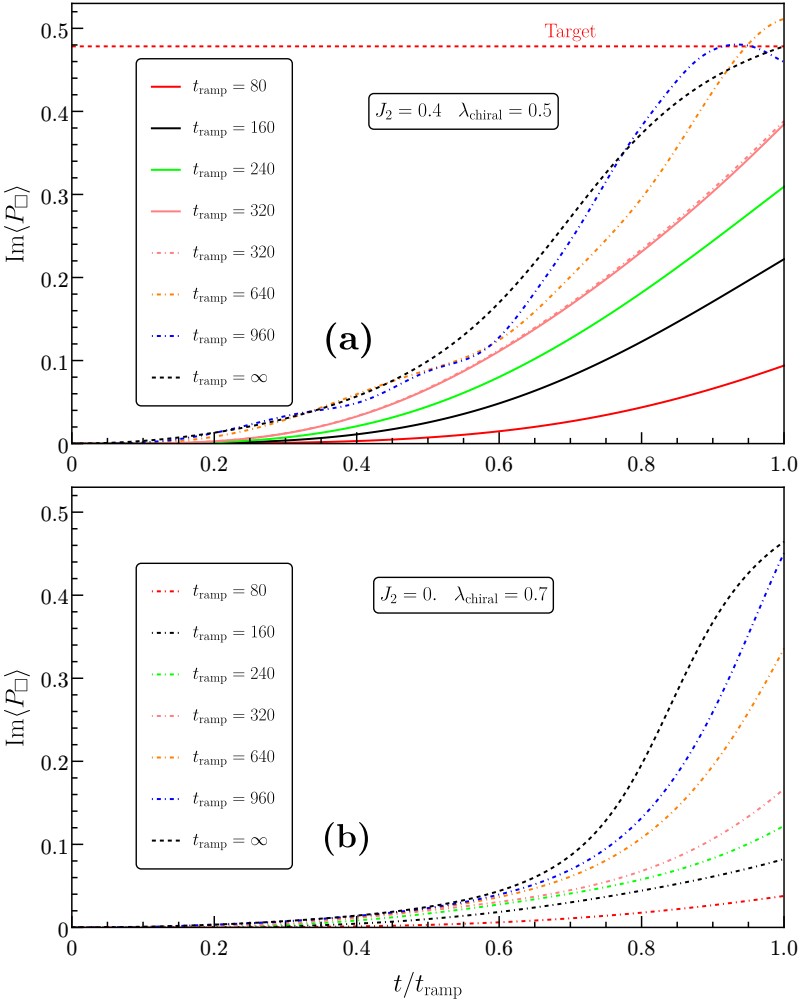

Figure 5: Plaquette chirality vs reduced time $t/t_{\text{ramp}}$ for adiabatic ramping of the drive using different values of $t_{\text{ramp}}$, computed on a $4 \times 4$ torus. (a) The RVB state is used as initial state, approximating the GS of the frustrated Heisenberg quantum antiferromagnet $H_0$ defined by $J_1 = 1$ and $J_2 = 0.4$. $\kappa$ is given by Eq. 11 with $\lambda_{\text{chiral}} = 0.5$. All time-dependent computations (shown as continuous curves) are done with $N_\tau = 16$ intervals in each drive period. (b) The GS of the Heisenberg quantum antiferromagnet $H_0$ ($J_1 = 1$, $J_2 = 0$) is used as initial state. $\kappa$ is given by Eq. 11 with $\lambda_{\text{chiral}} = 0.7$. (a,b) Results using the (weakly time-dependent) effective Floquet Hamiltonian of Eq. 10 are shown as dashed-dotted curves. The (ideal) adiabatic limits are shown by green dotted lines.

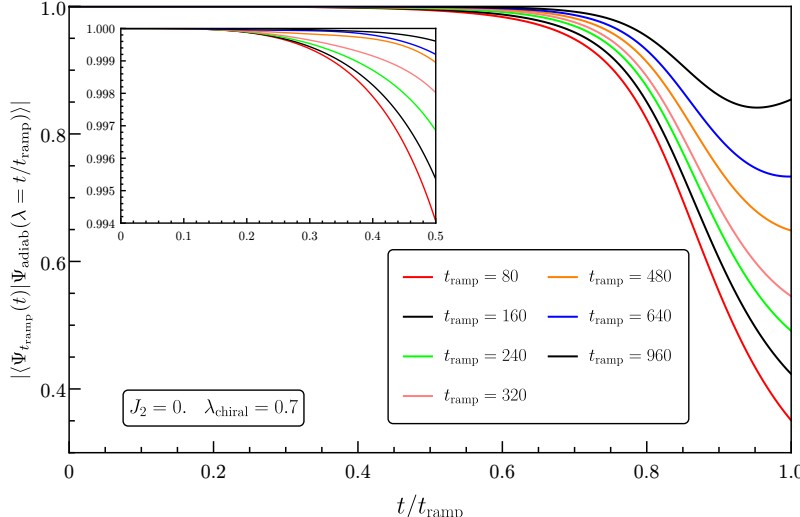

Figure 6: Overlap $|\langle \Psi_{t_{\text{ramp}}}(t)|\Psi_{\text{adiab}}(\lambda = t/t_{\text{ramp}})\rangle|$ between the time-evolved state and the adiabatic GS computed at $\lambda = t/t_{\text{ramp}}$, plotted versus $t/t_{\text{ramp}}$. Here the time-evolved state is obtained using the simplified Floquet dynamics with $J_1 = 1$, $J_2 = 0$ and $\lambda_{\text{chiral}} = 0.7$ and various finite ramp times $t_{\text{ramp}}$ up to $t_{\text{ramp}} = 960$ are considered. The GS of $H_0$ is taken as initial state. The inset is a zoom of the same quantity for $t/t_{\text{ramp}} \leq 0.5$.

Floquet Hamiltonian $H_F(t)$ gives very similar results as seen in Fig. 5(a). In particular, the overlap $|\langle \Psi_{\text{Floquet}}(t)|\Psi_{\text{drive}}(t)\rangle|$ between the two final states at $t = t_{\text{ramp}}$ is 0.99908, very close to 1. The effective Floquet dynamics allows to use much longer Trotter steps, typically of order 0.5, compared to $T_{\text{drive}}/N_\tau = 6.25 \times 10^{-3}$ for the full dynamics, and is therefore a much cheaper alternative for longer times. Hereafter, further results will then be obtained using Floquet dynamics, showing no appreciable difference on the plots and performed easily up to $t_{\text{ramp}} = 960$ (i.e. around 10 000 drive time periods!). For $t_{\text{ramp}} > 500$, while approaching the adiabatic limit, one starts to see oscillations in the time evolution of the chirality which are attributed to the deviation of the initial RVB state from the true GS of $H_0$.

To confirm the origin of the spurious oscillations found above, we consider the simple quantum Heisenberg antiferromagnet, setting $J_1 = 1$, $J_2 = 0$ in $H_0$, and use its GS as initial state, instead of the RVB state. Note that, for this new choice of $H_0$, the overlap between the RVB state and the true GS is only 0.9256 on the $4 \times 4$ torus and, hence, the RVB is no longer an appropriate choice for the initial state. Results are shown in Fig. 5(b), showing a smooth evolution towards the adiabatic limit when increasing $t_{\text{ramp}}$, even at large $t_{\text{ramp}}$, in contrast to Fig. 5(a).

## 3.4 Fidelities of the final CSL state

The results shown in the previous sections strongly suggest that the ramp setup can give efficient implementations of the CSL state. Here we try to analyse more quantitatively the properties of the final state.

### 3.4.1 Comparison with the adiabatic limit

First, we show in Fig. 6, for various values of $t_{\text{ramp}}$, the overlap of the time-evolved state $|\Psi_{t_{\text{ramp}}}(t)\rangle$ with the adiabatic GS $|\Psi_{\text{adiab}}(\lambda)\rangle$, computed at $\lambda = t/t_{\text{ramp}}$, as a function of the reduced time $t/t_{\text{ramp}}$. As expected, we observe that, for a given ramp time, the fidelity dete-

riorates as a function of time. However, for a fixed ratio $\lambda = t/t_{\text{ramp}}$ defining a target (chiral) state $|\Psi_{\text{adiab}}(\lambda)\rangle$ GS of a target Hamiltonian where $\lambda_{\text{chiral}}$ is replaced by $\lambda^2 \lambda_{\text{chiral}}$, the fidelity quickly increases with $t_{\text{ramp}}$. For $\lambda = 1$, corresponding to $\lambda_{\text{chiral}} = 0.7$ in $H_{\text{target}}$, a fidelity of $\sim 0.854$ can be obtained using $t_{\text{ramp}} \simeq 1000$. For $\lambda = 0.8$, corresponding to $\lambda_{\text{chiral}} \simeq 0.45$, a fidelity of $\sim 0.953$ can be reached with the same value of $t_{\text{ramp}}$, but after a shorter time $\lambda\, t_{\text{ramp}} \simeq 800$.

### 3.4.2 Comparison with chiral PEPS

We expect the final state to be in a CSL phase, in the close neighborhood of the GS of the target Hamiltonian. The later has been shown to be represented faithfully by chiral PEPS ansatze [44, 56]. It is therefore instructive to compare our final states prepared at $t = t_{\text{ramp}}$ with optimized PEPS. We shall consider fully symmetric, i.e. SU(2) and translationally invariant, PEPS with bond virtual spaces $\nu = 0 \oplus 1/2$ ($D = 3$), $\nu = 0 \oplus 1/2 \oplus 1/2$ ($D = 5$) and $\nu = 0 \oplus 1/2 \oplus 1$ ($D = 6$). Within these symmetric PEPS manifolds, restricted $A_1 + iA_2$ chiral sub-manifolds (with same bond dimensions) are designed to represent our CSL (see Appendix A for more details). While generic manifolds possess ($n_{\text{tensors}} - 1$) *complex* independent parameters, where $n_{\text{tensors}}$ is the total number of $A_1$ and $A_2$ tensors (see Table 1 for numbers), the chiral PEPS manifolds (with a fixed relative phase between the $A_1$ and $A_2$ tensors) have only half degrees of freedom i.e. ($n_{\text{tensors}} - 1$) *real* parameters to be optimized.

    As shown in Fig. 7, as expected the $D = 3$ PEPS cannot accommodate very well, for increasing $t_{\text{ramp}}$, the increase of entanglement of $|\Psi_{t_{\text{ramp}}}\rangle$ signaled by the rapid decrease of the fidelity w.r.t. the initial state. However, one already sees that the performance of the chiral PEPS (with only two variational parameters) becomes comparable to the generic (symmetric) PEPS for $t_{\text{ramp}} > 250$, suggesting a crossover towards the CSL phase. The extension of the chiral PEPS to $D = 5$ or $D = 6$ unfortunately does not improve the overlaps significantly (see Table 1) but a similar trend is found as shown in Fig. 7(b) for $D = 6$.

## 4 Discussions and conclusions

We first discuss possible experimental realisations of our setup, how to realize the fast drive and how to prepare the initial state.

### 4.1 Realizing the drive Hamiltonian

The Heisenberg interactions in Eq. 5 can be realized with cold atoms loaded on an optical lattice [61], for which a microscopic Hubbard-like description applies [62,63]. The excellent isolation of the optical lattice from its environment enables to realize ideal unitary dynamics. We assume one atom per site (on average) and a large on-site repulsion $U$ to be in the Mott insulating phase [64], and different hopping terms $t_\nu(t)$ on the four NN bonds $\nu = a, b, c, d$ of Fig. 1. Our setup is relatively simple as it involves only the modulation in time and space of the NN (effective) Heisenberg couplings. A (selective) modulation in time of the Heisenberg couplings $J_\nu(t)$ of $H_\nu$ can be realized via a (selective) modulation of the hoppings between sites $t_\nu(t)$, i.e. changing the tunneling barriers of the optical lattice appropriately. Note that this procedure is conceptually different from the synthetic gauge field approach [18], in which the chiral term would appear in the Mott insulating phase only in fourth-order of the (complex) hopping $t$, i.e. $\lambda_{\text{chiral}} \sim t^4/U^3$. However, in fermionic Hubbard systems the Heisenberg couplings can only be antiferromagnetic ($J_\nu = 4(t_\nu(t))^2/U > 0$), not ferromagnetic ($J_\nu < 0$) and,

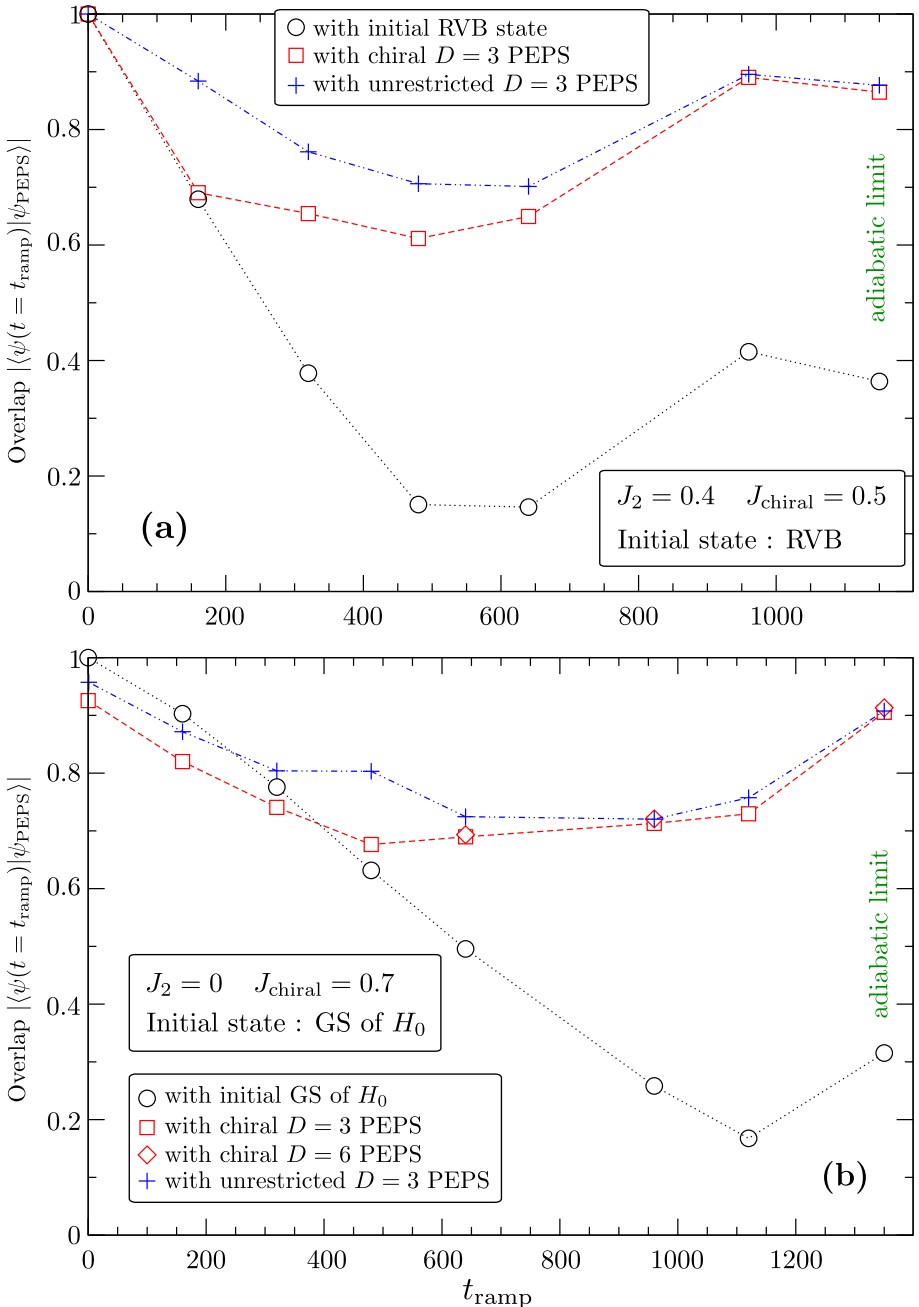

Figure 7: Overlap $|\langle \Psi_{t_{ramp}} | \Psi_{PEPS} \rangle|$ between the time-evolved state at $t = t_{ramp}$ and the best optimized PEPS states, plotted versus $t_{ramp}$. (a) $J_1 = 1$, $J_2 = 0.4$ and $\lambda_{chiral} = 0.5$, and NN RVB state as initial state; (b) $J_1 = 1$, $J_2 = 0$ and $\lambda_{chiral} = 0.7$, and GS of $H_0$ as initial state. Different symbols correspond to different PEPS manifold, as indicated in the legend. The overlap $|\langle \Psi_{t_{ramp}} | \Psi_{init} \rangle|$ with the initial state is shown as a reference. The adiabatic limit expected for $t_{ramp} \to \infty$ is shown on the right of the picture.

strictly speaking, the single-harmonic modulations of (5) cannot be realized.[2] A more practical alternative would then be to use, on the optical lattice, two-component bosons [65,66] which (i) are more easily brought to low temperatures and (ii) are more versatile than fermions to realize alternating antiferromagnetic and ferromagnetic couplings, tuning the $U_{\uparrow\uparrow} = U_{\downarrow\downarrow}$ and $U_{\uparrow\downarrow}$ on-site Hubbard interactions.

It is interesting to mention that a simplified drive sequence consists in replacing the sinusoidal modulation by a square wave i.e.

$$\tilde{H}_{\text{drive}}(t) = H_x, \quad \text{for} \quad t \in \left[-\frac{T}{4}, \frac{T}{4}\right],$$

$$\tilde{H}_{\text{drive}}(t) = H_y, \quad \text{for} \quad t \in \left[0, \frac{T}{2}\right],$$

$$\tilde{H}_{\text{drive}}(t) = -H_x, \quad \text{for} \quad t \in \left[\frac{T}{4}, \frac{3T}{4}\right],$$

$$\tilde{H}_{\text{drive}}(t) = -H_y, \quad \text{for} \quad t \in \left[\frac{T}{2}, T\right]. \tag{17}$$

This new type of drive sequence introduces higher harmonics $\frac{2}{n\pi}(\exp(in\omega t) \pm c.c.)$ in the amplitude modulations of $H_x$ and $iH_y$ which modify the prefactor $J_F$ in Eq. 7 into

$$\tilde{J}_F = \frac{14\zeta(3)}{\pi^2} J_F \simeq 1.72 J_F, \tag{18}$$

and the chiral term in Eqs. 7 and 10 accordingly. Note also the emergence of chiral terms in the (targeted) Floquet effective Hamiltonian at order $\mathcal{O}(\omega^{-2})$ instead of $\mathcal{O}(\omega^{-3})$. Nevertheless, we believe the main ideas/results developed in this work are still valid qualitatively in the case of (17). Finally, we would like to mention that Rydberg atom arrays can also emulate quantum spin systems [67], but keeping SU(2) spin rotational invariance might be challenging. Nevertheless, it would be interesting to investigate whether our setup can be exported to the case of anisotropic XXZ dipolar spin interactions.

## 4.2 Preparation of the initial state

Spin liquids of the RVB type can be realized on cold atom platforms i.e. in Mott insulating ultracold bosons [68]. Using a two-component one-dimensional Bose-Hubbard system, a highly entangled Heisenberg antiferromagnet with SU(2) symmetry has been experimentally realized [69]. We believe that this promising approach could be extended to two dimensions. Note however that, as shown by our studies, it is important that the initial state in the ramp process is as close as possible to the GS of the Hamiltonian at $t = 0$ i.e. $H(0) = H_0$. Using cold atoms on an optical square lattice, the Mott phase can be realized on fermionic systems of hundred of sites [70,71] with magnetic correlation lengths of several lattice spacings [72]. Similarly, systems of two-component bosons could also be brought in Mott phases at large Hubbard repulsions $U_{\sigma\sigma'}$ [66]. Owing to rapid experimental progress in decreasing the temperature, we believe the quantum state of such systems may soon approach the (finite size) GS of a plain spin-1/2 NN quantum Heisenberg model (i.e. with $J_2 = 0$) and could serve as an initial state for the second type of ramping. Tweezer arrays of fermions have also been proposed [73], that could be used to reach the same goal. We point out that the precise nature of the initial state (e.g. spin liquid vs Néel state) is not completely settled on finite size systems which are always quantum-mechanically disordered (i.e. are in a global singlet state)

---

[2]A constant NN antiferromagnetic interaction $J$ cannot be added to all bonds since it will contribute to the constant $H_0$ part of $H_F$.

and possess a small gap in their many-body spectrum. In other words, the distinction between different phases is not sharp on finite size systems.

## 4.3 Final remarks

In summary, we have studied different protocols to prepare a (finite size) system of quantum spin-1/2 on a square lattice into a spin-1/2 chiral spin liquid phase. We start from a simple low-entangled (SU(2)-symmetric) initial state like a (non-chiral) spin liquid (e.g. a RVB spin liquid) or the (finite size) GS of a simple NN Heisenberg quantum antiferromagnet which, we believe, could (or will soon) be realized experimentally in platforms of cold atoms on an optical lattice. A simple drive involving dephased periodic modulations of four types of NN bonds is proposed. We focus on the high-frequency/large amplitude (strong drive) regime for which an effective Floquet dynamics is shown to give very accurate results. We argue that our goal cannot be realized by quenching the drive suddenly, but rather by a semi-adiabatic ramping. Quantitative results are given for a periodic $4 \times 4$ system. In that case, we show that the (finite size) CSL ground state of a relevant target Hamiltonian can be reached with high fidelity typically within less than 10 thousands drive oscillations, still beyond current experimental possibilities. However, high-fidelity counterdiabatic protocols – i.e. far away from the adiabatic limit – respecting experimental limitations (like the limited coherence times) have been proposed [74] and may be adapted to our setup.

We would like to emphasize again that it is essential that the system is kept finite and we believe the thermodynamic limit cannot be taken right away for two reasons: first, heating will occur when the many-body Floquet spectrum (expanding with system size as $N/\omega$) will reach the boundary of the Floquet-Brillouin quasi-energy zone of extension $\omega$. This simple argument gives a minimum critical frequency which diverges as $\sqrt{N}$ for increasing system size; secondly, another issue is the vanishing of the finite size gap in the thermodynamic limit that may lead to a diverging ramp time.

Recent experiments are now performed on systems with hundred sites or even more, and with a box-like confining potential (corresponding theoretically to open boundary conditions). The use of periodic boundary conditions in our system was in fact dictated by convenience, translation invariance making calculations simpler. However, we believe no conceptual difference is to be expected for open boundaries. The case of a harmonic confining potential is more tedious and left for future investigations. In any case, the system size $N$ is an important relevant parameter. Indeed, in the transition region between the non-chiral initial phase and the CSL, the gap should reach a minimum value, typically of order $1/N$, or a power of $1/N$. Hence, the minimum ramp time to reach a good fidelity should qualitatively scale like $N$, which could be a limitation for $N$ of a few hundred sites. However we believe that preparation times could be drastically reduced using quantum control algorithms, an interesting issue left for future studies.

Our computation has been performed on the square lattice but similar setups can also be realized in frustrated 2D lattices like kagome or triangular lattices which can be obtained experimentally [75]. In that case, dephased (fast) modulations of 3 or 6 groups of equivalent NN bonds can be performed leading to chiral permutations on triangles or hexagons. The resulting topological CSL state and its anyonic excitations may be used for topological quantum computations following Kitaev's proposal [51].

We also note that we have restricted ourselves to the high-frequency regime. However, in Ref. [58] it is argued that there exists novel non-equilibrium chiral phase in the limit of a slow drive. This was then generalized to a Kitaev type model [76]. Investigations of our model at lower frequency is of great interest and is currently undertaken looking for a similar phase in an SU(2) symmetric model.

Lastly, we note that the implementation of a dimer spin liquid state (of the toric code universality class) has been proposed on the kagome lattice by using Rydberg atom arrays [67]. It would then be interesting to investigate whether a chiral dimer liquid (CDL) could be realized, starting from such an initial state, by Floquet engineering involving dimer degrees of freedom (represented by excited Rydberg atoms in the blockade regime) instead of the original spin-1/2 degrees of freedom considered in this work.

## A  PEPS representation of spin liquids

PEPS are defined as a tensor network of local onsite tensors of size $dD^4$ where $d = 2$ is the local physical Hilbert space dimension (spin 1/2) and $D$ is the bond dimension i.e. the number of virtual states on each tensor leg. By contracting the tensors w.r.t. the bond indices, one generates an entangled ansatz (whose entanglement entropy per bond is limited by $\ln D$ and, hence, controlled by $D$). PEPS offer an extremely efficient variational scheme to address local Hamiltonians [77]. In addition, PEPS can encode the topological nature of the state and the physics of the edge modes via the PEPS bulk-edge correspondence theorem [78, 79]. In the infinite-PEPS (iPEPS) algorithm [42] infinite-size systems can be handled. Note that PEPS fail to describe the rapid increase of entanglement entropy (e.g. in the case of a quench, see Ref. [80]) but should still be relevant in the case of an adiabatic (or slow) ramp which leads to a limited growth of entanglement entropy.

Both the initial state and the Floquet Hamiltonian governing the *effective* dynamics of our problem are SU(2) symmetric and transform according to the trivial representation of the square lattice space group.[3] It is therefore relevant to enforce such symmetries at every step of our scheme. A symmetric spin liquid on the square lattice is simply obtained by choosing a $C_{4v}$-symmetric site tensor placed uniformly on every lattice sites. As explained in details in Ref. [81], the continuous SU(2) of PEPS can be implemented at the level of local tensors by imposing that the $D$-dimensional subspace $v$ of each virtual leg is a direct sum of SU(2) irreducible representations. Working with SU(2)-symmetric PEPS allows to greatly reduce the number of parameters describing the PEPS family and is a major advantage for computations based on optimization.

The NN RVB state [37] which consists of an equal weight superposition of all NN singlet (valence bond) coverings[4] has a simple representation using $v = 0 \oplus 1/2$ virtual bonds ($D = 3$) [47, 48]. In fact, the $D = 3$ symmetric PEPS manifold is represented by two linearly independent fully symmetric ($A_1$) real site tensors (describing the fusion of either one or three virtual spin-1/2 into a physical spin-1/2) and generically describes gapped $\mathbb{Z}_2$ topological RVB spin liquids which include (thanks to "teleportation") longer range singlet bonds [49]. The critical NN RVB state is therefore a special point in this two-parameter manifold [50], with exponentially large correlation lengths in its neighborhood [49, 50].

The simplest PEPS representation of the (Abelian) spin-1/2 CSL – of the Kalmeyer-Laughlin type [2] – is in fact possible within the $v = 0 \oplus 1/2$ virtual space ($D = 3$) manifold by adding a third site tensor of different $A_2$ orbital symmetry with a pure-imaginary amplitude, resulting into a $A_1 + iA_2$ site tensor [43, 44]. Such an ansatz breaks time-reversal T ($i \rightarrow -i$) and parity P ($A_2 \rightarrow -A_2$) while being invariant under their product PT, as expected for a CSL. Also, despite its simplicity, it gives a faithful description of the chiral edge modes – a key feature of the CSL – following closely the prediction of a SU(2)$_1$ Wess-Zumino-Witten (WZW) conformal field

---

[3]Note that the time-dependent drive (5) breaks the $C_{4v}$ point group symmetry causing small (negligible) lattice asymmetry of the time-evolved state at short time $t < 1/J$.

[4]On a bipartite lattice, NN singlets are defined in the bosonic representation oriented from one sublattice to the other.

Table 1: Number of $A_1$ and $A_2$ tensors and number of variational parameters (fixing one tensor amplitude) used to construct the chiral PEPS, as a function of the bond dimension. The last lines show the corresponding overlaps with the two adiabatic GS considered in this work.

| D | 3 | 5 | 6 |
|---|---|---|---|
| no of $A_1$ tensors | 2 | 10 | 11 |
| no of $A_2$ tensors | 1 | 8 | 8 |
| no of variational parameters | 2 | 17 | 18 |
| $\left|\left\langle \Psi_{\text{adiab}}^{(a)} \middle| \Psi_{\text{PEPS}} \right\rangle\right|$ | 0.86470 | 0.86648 | |
| $\left|\left\langle \Psi_{\text{adiab}}^{(b)} \middle| \Psi_{\text{PEPS}} \right\rangle\right|$ | 0.90525 | 0.91256 | 0.91319 |

theory (CFT). Moreover, the CSL ground-state [82] of the target Hamiltonian (3) has been successfully studied using similar PEPS [53,56]. Increasing the bond dimension from $D = 3$ to $D = 5$ ($\nu = 0 \oplus 1/2 \oplus 1/2$) or $D = 6$ ($\nu = 0 \oplus 1/2 \oplus 1$) provides a small improvement of the variational energy. We show in Table 1 the optimal PEPS overlaps on a $4 \times 4$ torus with the adiabatic GS considered in this work.

# B Derivation of the Floquet effective Hamiltonian

## B.1 $H_0 = 0$ case

We start from the expression of the Floquet Hamiltonian at order $1/\omega$ – see Eq. (42) in [41] – decomposing $H_{\text{drive}}$ as $H_1 \exp(i\omega t) + H_{-1} \exp(-i\omega t)$,

$$
\begin{aligned}
H_F^0 &= \frac{1}{\omega}[H_1, H_{-1}] \\
&= \frac{i}{2\omega}[H_x, H_y] \\
&= 2J_F\, i[h_x, h_y],
\end{aligned}
\tag{19}
$$

where

$$
\begin{aligned}
h_x &= H_x/J = \sum_{i\in A}(\mathbf{S}_i \cdot \mathbf{S}_{i+e_x} - \mathbf{S}_i \cdot \mathbf{S}_{i-e_x}), \\
h_y &= H_y/J = \sum_{i\in A}(\mathbf{S}_i \cdot \mathbf{S}_{i+e_y} - \mathbf{S}_i \cdot \mathbf{S}_{i-e_y}).
\end{aligned}
\tag{20}
$$

It is interesting to note that, in this simple case of an harmonic drive alone, corrections only appear at order $1/\omega^3$. One can then expand the commutator as $[h_x, h_y] = \sum_{i\in A}(C_i + C_i')$ where,

$$
\begin{aligned}
C_i &= \left(S_{i+e_x}^\alpha - S_{i-e_x}^\alpha\right)[S_i^\alpha, S_i^\beta]\left(S_{i+e_y}^\beta - S_{i-e_y}^\beta\right), \\
C_i' &= \left(S_i^\alpha - S_{i+2e_x}^\alpha\right)[S_{i+e_x}^\alpha, S_{i+e_x}^\beta]\left(S_{i+e_x-e_y}^\beta - S_{i+e_x+e_y}^\beta\right).
\end{aligned}
\tag{21}
$$

Using $[S_i^\alpha, S_i^\beta] = i\epsilon_{\alpha\beta\gamma}S_i^\gamma$ we get,

$$
C_i = i\mathbf{S}_i \cdot (\mathbf{S}_{i+e_x} \times \mathbf{S}_{i+e_y}) + i\mathbf{S}_i \cdot (\mathbf{S}_{i+e_y} \times \mathbf{S}_{i-e_x}) + i\mathbf{S}_i \cdot (\mathbf{S}_{i-e_x} \times \mathbf{S}_{i-e_y}) + i\mathbf{S}_i \cdot (\mathbf{S}_{i-e_y} \times \mathbf{S}_{i+e_x}).
\tag{22}
$$

After the transformation $i + e_x \to j$, $C'_i \to C_j$, we obtain for $j \in B$,

$$
\begin{aligned}
C_j &= (S^\alpha_{j-e_x} - S^\alpha_{j+e_x})\big[S^\alpha_j, S^\beta_j\big]\big(S^\beta_{j-e_y} - S^\beta_{j+e_y}\big) \\
&= i\mathbf{S}_j \cdot (\mathbf{S}_{j+e_x} \times \mathbf{S}_{j+e_y}) + i\mathbf{S}_j \cdot (\mathbf{S}_{j+e_y} \times \mathbf{S}_{j-e_x}) + i\mathbf{S}_j \cdot (\mathbf{S}_{j-e_x} \times \mathbf{S}_{j-e_y}) + i\mathbf{S}_j \cdot (\mathbf{S}_{j-e_y} \times \mathbf{S}_{j+e_x}).
\end{aligned}
\tag{23}
$$

Combining the last two equations, we get

$$
\begin{aligned}
i[h_x, h_y] &= -\sum_{i \in A,B} \Big\{ \mathbf{S}_i \cdot (\mathbf{S}_{i+e_x} \times \mathbf{S}_{i+e_y}) + \mathbf{S}_i \cdot (\mathbf{S}_{i+e_y} \times \mathbf{S}_{i-e_x}) \\
&\qquad\qquad + \mathbf{S}_i \cdot (\mathbf{S}_{i-e_x} \times \mathbf{S}_{i-e_y}) + \mathbf{S}_i \cdot (\mathbf{S}_{i-e_y} \times \mathbf{S}_{i+e_x}) \Big\} \\
&= -\sum_{\square} \Big\{ \mathbf{S}_i \cdot (\mathbf{S}_{i+e_x} \times \mathbf{S}_{i+e_y}) + \mathbf{S}_{i+e_x} \cdot (\mathbf{S}_{i+e_x+e_y} \times \mathbf{S}_i) \\
&\qquad\qquad + \mathbf{S}_{i+e_x+e_y} \cdot (\mathbf{S}_{i+e_y} \times \mathbf{S}_{i+e_x}) + \mathbf{S}_{i+e_y} \cdot (\mathbf{S}_i \times \mathbf{S}_{i+e_x+e_y}) \Big\} \\
&= \frac{1}{2} \sum_{\square} i(P_{ijkl} - P^{-1}_{ijkl}),
\end{aligned}
\tag{24}
$$

where the sum over all sites has been rewritten as a sum over all (oriented) plaquettes $(i, j, k, l) \equiv (i, i + e_y, i + e_x + e_y, i + e_x)$ of the square lattice. Substituting Eq. 24 into Eq. 19 gives the expression of the Floquet Hamiltonian (7) of the main text.

## B.2 Finite constant $H_0$

When a small constant term $\kappa H_0$, $\kappa \ll 1$, is present in the Hamiltonian a small correction to the Floquet Hamiltonian appears [41], in addition to the constant term itself,

$$
\begin{aligned}
\delta H_F &= \frac{1}{\omega}(-[H_1, H_0] + [H_{-1}, H_0]) \\
&= \kappa \frac{J J_1}{\omega} i[h_y, h_0],
\end{aligned}
\tag{25}
$$

where $h_0 = H_0/J_1$ is the dimensionless uniform (frustrated) Heisenberg model. By expanding $h_y$ and $h_0$ one obtains a sum of chiral terms on 2 types of triangles, $(i, i + e_x, i + e_y)$ and $(i, i + e_x, i + 2e_x + e_y)$, with alternating chiralities. Since $\kappa$ is of order $J/\omega$, the amplitude of $\delta H_F$ is of order $\frac{J_1 J^2}{\omega^2}$ i.e. of order $\omega^{-2}$ and, hence, has been neglected in (10).

Note however that, in the case of a ramp, $\delta H_F$ has to be multiplied by $\lambda(t)$, in contrast to $H^0_F$ multiplied by $\lambda^2(t)$ in (10). Therefore at very short time, $\frac{t}{t_{\text{ramp}}} < \frac{J_1}{\omega}$, $\delta H_F$ may *a priori* play some role. However, we have found numerically, that the Floquet dynamics obtained with (10) was providing very accurate results compared to those obtained with the full time-dependent Hamiltonian.

# C  Derivation of the kick operator

In the stroboscopic picture, the time-evolved state can be written as

$$
|\Psi(t)\rangle = \exp(-iK(t))\exp(-iH_F t)|\Psi_0\rangle,
$$

where $K(t)$ is the Hermitian periodic kick operator (of period $T_{\text{drive}}$). The unitary operator $\exp(-iK(t))$ corresponds physically to a change of reference frame (via a change of basis).

At all stroboscopic times $t = nT_{\text{drive}}$, $n \in \mathbb{N}$, we have $K(t) = 0$ and $\exp(-iK(t)) = \mathbb{I}_d$. Using Eq. (44) of Ref. [41], we get the leading high-frequency term:

$$
\begin{aligned}
K(t) &= \frac{1}{i\omega} \left[ H_1(e^{i\omega t} - 1) - H_{-1}(e^{-i\omega t} - 1) \right] \\
&= \frac{J}{\omega} \left[ (1 - \cos(\omega t)) h_y + \sin(\omega t) h_x \right].
\end{aligned}
\tag{26}
$$

Because of the small prefactor in the expression of $K(t)$, the unitary remains close to the identity. Most observables are therefore weakly affected during the micromotion w.r.t. to their behaviors under the effective Floquet dynamics. This has been confirmed numerically.

# Acknowledgments

D.P. acknowledges inspiring discussions with Mari-Carmen Banŭls, Antoine Browaeys, Thierry Giamarchi, Nathan Goldman, David Guery-Odelin, Vedika Khemani, Michael Kolodrubetz, Thierry Lahaye, Joel Moore, Sen Niu and Luca Tagliacozzo. This work was granted access to the HPC resources of CALMIP center under allocations 2023-P1231 and 2024-P1231.

**Funding information** D.P. acknowledges support from the TNTOP ANR-18-CE30-0026-01 grant awarded by the French Research Council.

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
