# Peer review of "Quantum state preparation of topological chiral spin liquids via Floquet engineering"

_SciPost Physics, doi:SciPost Phys. 17, 011 (2024)_

## Round 2 · Referee Report · Marin Bukov (Referee 1) · 2024-3-29

Strengths

  • timely study

  • touches upon various aspects of nonequilibrium dynamics, with particular focus on Hamiltonian engineering and state preparation of chiral spin liquid states

  • comparison between theory and experiment shows results are solid and justified.

Weaknesses

  • a few statements require clarification

Report

The paper "Quantum state preparation of topological chiral spin liquids via Floquet engineering" by Mambrini and Poilblanc is an interesting and timely study, touching upon various aspects of nonequilibrium dynamics, with particular focus on Hamiltonian engineering and state preparation of chiral spin liquid states.

The paper is interesting and can be published once the authors have considered the following points:

Sec 2:

  • Eq(3); it would be helpful if the authors define explicitly the permutation operator in terms of the spin degrees of freedom (I now see that you sort of have it in Eq(24) but an explicit reference will be useful)

  • Sec 2.3, "Note that such a monochromatic drive may not be easy to implement on experimental cold atom platforms": I think there is no difficulty with the dephased monochromatic drive itself; the spin-spin interactions would be the real challenge, see e.g., https://www.nature.com/articles/s41567-020-0949-y

  • Sec 2.4: before Eq(6), when the authors introduce stroboscopic times, there appear integers $p$, $n$, times $t$, $t_0$, etc,; what's written is not wrong, but I think it can be simplified by removing the unnecessary info for the benefit of the readers.

  • after Eq(7), "It is interesting that the next order of the expansion is O(1/ω3) so that we expect HF0 to capture accurately the stroboscopic motion.": this is not obvious to me; can the authors argue why there are no 1/omega^2 terms here?

  • if I'm not mistaken, you might be able to compute the Floquet Hamiltonian to Eq(6) alone exactly, as follows:

1) Let $L_z$ be the generator of continuous rotations about the out-of-lattice-plane axis, i.e., the operator that rotates $H_x$ into $H_y$: $\exp(-i \pi/2 L_z) H_x \exp(+i \pi/2 L_z) = H_y$; I think, $\exp(-i \pi/2 L_z)$ is related to the permutation operator $P_{i,j,k,l}$ but we need the infinitesimal rotation generator. Not sure how crazily non-local (in terms of the spin operators) $L_z$ is, though.

2) consider now a rotation about the angle $\alpha = \omega t$: in a co-rotating frame defined by the rotation $V(t) = \exp(-i \omega t L_z)$, the circular drive will freeze and point, let's say, along $H_x$.

3) taking into account the Galilean term due to the frame transformation, in the co-rotating frame the Hamiltonian will read as $H' = \omega L_z + H_x$ (no liability regarding the correctness of signs).

This business is very similar to the two-level system in a circular drive, just the operators involved are quite a bit more complicated. To relate $H'$ to $H_F$, you may need to do extra work, especially if the spectrum of $L_z$ is not commensurate, see Sec 2.3 of Ref [39]. In the language of $H'$, to make the chiral structure manifest, you'd have to consider the "symmetric gauge", see discussion in Sec 3.3 of Ref [39].

Sec 3:

  • Sec 3.2.1: "Since the effective Floquet dynamics is expected to be very accurate, one can use it to investigate the physics in the t → ∞ limit": this is dangerous b/c the approximate effective Hamiltonian computed from the Magnus expansion does not capture heating effects. The system is expected to heat up to an infinite temperature state in the t → ∞ limit for any finite $\kappa>0$, see e.g., Ref [33]. However, what's physically relevant in practice is the prethermal plateau regime which occurs at finite time (and whose lifetime can be controlled by increasing the drive frequency).

  • do PEPS/chiral PEPS offer a controlled approximation to CSL states in large systems? What about iPEPS? Are the deviations from unity in the fidelities reported in Table 1 due to the failure of the adiabatic ramp or due to the PEPS ansatz not reaching the ground state?

App A:

  • "PEPS offer an extremely efficient variational scheme to address local Hamiltonians": it's not clear which features/properties are being addressed
  • it's worthwhile to mention in one sentence the drawbacks of PEPS as well, or the regime of applicability

App C:

  • "The unitary operator exp(−iK(t)) corresponds physically to a change of basis": time-dependent unitaries correspond rather to a change of reference frame; when evaluated at some fixed t_0, one can think of them as giving kicks to the state (i.e., changing the basis). In this sense, Floquet's theorem is a statement about the existence of a reference frame, where the dynamics is governed by a static Hamiltonian H_F at all times (not only stroboscopically).

Figs:

  • Fig 2:

a) I find it a bit confusing that the micromotion in inset (c) seems to oscillate around the black-squares curve; however the two curves are w.r.t. different x-axes (black squares go from times ~25 to 40, while grey micromotion curve goes from times 30.1 to 30.5). It might be better to show inset (c) as a separate panel of the figure to avoid this confusion.

b) regarding errors caused by Trotterization: I think it should be possible to evolve the state using a Runge-Kutta solver on a 4x4 patch.

  • Fig 3: is $k$ the same integer as $p$? If yes, better use $p$.

References:

  • non-abelian anyons have recently been observed on Honeywell's trapped ion quantum computer: https://www.nature.com/articles/s41586-023-06934-4

  • the authors might be interested in a recent preprint, where we discuss various aspects of state preparation under strong Floquet drives, and in particular how to speed up adiabatic ramps: https://arxiv.org/abs/2310.02728

Typos:

abstract: remarquable --> remarkable after Eq(1) [+ other instances]: "significantly smaller that the smallest": that --> than before Eq(2): Krylov’s --> Krylov Eq(2): summation index $m$ missing in sum subscript Sec 2.2, first paragraph, "It can be view as": view --> viewed Sec 3.1, penultimate paragraph, "p fixed": p --> $p$ Sec 3.2, first paragraph, "is suddenly switch on", switch --> switched Sec 3.4.2, last paragraph, "As shown in Figs. 7": Figs --> Fig Sec 4.3: "We argue that our goal cannot be realized by branching the drive suddenly": branching --> quenching Sec 4.3: "the minimum ramp time to reach a good fidelity should qualitatively scales like N": scales --> scale

Requested changes

see report above

  • validity: top
  • significance: high
  • originality: top
  • clarity: top
  • formatting: perfect
  • grammar: excellent

Author:  Matthieu Mambrini  on 2024-04-29  [id 4450]

(in reply to Report 1 by Marin Bukov on 2024-03-29)

We thank Prof. Marin Bukov very much for is insightful reading of the paper. We have made some revisions and added some references following his comments. We believe it has improved the quality and clarity of the paper. Below we answer point by point and describe the changes made.

Section 2

Referee Eq(3); it would be helpful if the authors define explicitly the permutation operator in terms of the spin degrees of freedom (I now see that you sort of have it in Eq(24) but an explicit reference will be useful).

Authors A sentence has been added to make the definition of $P_{ijkl}$ explicit. The expression in terms of the spin degrees of freedom has now been included in the main text.

Referee Sec 2.3, "Note that such a monochromatic drive may not be easy to implement on experimental cold atom platforms": I think there is no difficulty with the dephased monochromatic drive itself; the spin-spin interactions would be the real challenge, see e.g., https://www.nature.com/articles/s41567-020-0949-y

Authors We thank the Referee for mentioning the reference. We have rephrased the short paragraph in Sec. 2.3 on the experimental implementation, taking into account the Referee comment.

Referee Sec 2.4: before Eq (6), when the authors introduce stroboscopic times, there appear integers p, n, times t, t0, etc,; what's written is not wrong, but I think it can be simplified by removing the unnecessary info for the benefit of the readers.

Authors Following the comment we have simplified the formulation (removing $t_0$, etc...).

Referee after Eq(7), "It is interesting that the next order of the expansion is $O(1/\omega^3)$ so that we expect HF0 to capture accurately the stroboscopic motion.": this is not obvious to me; can the authors argue why there are no $1/\omega^2$ terms here?

Authors Eq. (42) of Bukov et al. suggests that, in the absence of a time independent part $H_0=0$, the expansion involves only nested commutators containing the same number of $H_1$ and $H_{-1}$ Fourier components of the Hamiltonian, i.e. odd powers of $1/\omega$. We have added a note on this.

Referee If I'm not mistaken, you might be able to compute the Floquet Hamiltonian to Eq(6) alone exactly, as follows:

  1. Let $L_z$ be the generator of continuous rotations about the out-of-lattice-plane axis, i.e., the operator that rotates $H_x$ into $H_y$: $\exp(-i\pi/2L_z) H_x \exp(+i\pi/2 L_z )= H_y$; I think, $\exp(-i\pi / 2 L_z)$ is related to the permutation operator $P_{i,j,k,l}$ but we need the infinitesimal rotation generator. Not sure how crazily non-local (in terms of the spin operators) $L_z$ is, though.

  2. consider now a rotation about the angle $\alpha = \omega t$: in a co-rotating frame defined by the rotation $V(t)=\exp(-i \omega t L_z)$, the circular drive will freeze and point, let's say, along $H_x$.

  3. taking into account the Galilean term due to the frame transformation, in the co-rotating frame the Hamiltonian will read as $H'= \omega L_z+H_x$ (no liability regarding the correctness of signs).

This business is very similar to the two-level system in a circular drive, just the operators involved are quite a bit more complicated. To relate $H'$ to HF, you may need to do extra work, especially if the spectrum of $L_z$ is not commensurate, see Sec 2.3 of Ref [39]. In the language of $H'$, to make the chiral structure manifest, you'd have to consider the "symmetric gauge", see discussion in Sec 3.3 of Ref [39].

Authors We thank the Referee for his proposal. However, we believe that, away from the high-frequency limit, the Floquet Hamiltonian becomes more and more non local (including all sorts of chiral terms on larger and larger closed loops). It would be very surprising that tractable expressions could be obtained generically. In any case, we postpone this calculation to future studies.

Section 3

Referee Sec 3.2.1: "Since the effective Floquet dynamics is expected to be very accurate, one can use it to investigate the physics in the $t\rightarrow \infty$ limit": this is dangerous b/c the approximate effective Hamiltonian computed from the Magnus expansion does not capture heating effects. The system is expected to heat up to an infinite temperature state in the $t\rightarrow \infty$ limit for any finite $\kappa>0$, see e.g., Ref [33]. However, what's physically relevant in practice is the prethermal plateau regime which occurs at finite time (and whose lifetime can be controlled by increasing the drive frequency).

Authors We agree with the referee and have reformulated the text. We have added a comment on the role of the finite system size.

Appendix A

Referee "PEPS offer an extremely efficient variational scheme to address local Hamiltonians": it's not clear which features/properties are being addressed

Authors Beside providing a very good variational energy, PEPS can encode the topological nature of the state and the physics of the edge mode. Note and Refs. [75,76] added.

Referee it's worthwhile to mention in one sentence the drawbacks of PEPS as well, or the regime of applicability

Authors PEPS fail to describe the rapid increase of entanglement entropy (e.g. in the case of a quench, see Ref. [77]) but should still be relevant in the case of an adiabatic ramp.

Appendix C

Referee "The unitary operator $\exp(-iK(t))$ corresponds physically to a change of basis": time-dependent unitaries correspond rather to a change of reference frame; when evaluated at some fixed $t_0$, one can think of them as giving kicks to the state (i.e., changing the basis). In this sense, Floquet's theorem is a statement about the existence of a reference frame, where the dynamics is governed by a static Hamiltonian $H_F$ at all times (not only stroboscopically).

Authors We have changed "change of basis" into "change of reference frame (via a change of basis)".

Figure 2

Referee a) I find it a bit confusing that the micromotion in inset (c) seems to oscillate around the black-squares curve; however the two curves are w.r.t. different x-axes (black squares go from times $\sim$<!-- -->25 to 40, while grey micromotion curve goes from times 30.1 to 30.5). It might be better to show inset (c) as a separate panel of the figure to avoid this confusion.

Authors We have modified the figure to avoid the confusion mentioned by the Referee.

Referee b) regarding errors caused by Trotterization: I think it should be possible to evolve the state using a Runge-Kutta solver on a $4\times4$ patch.

Authors We have no expertise with such a technique but we have full control on the error introduced by the Trotterization scheme. By varying the Trotter step we can estimate the minimum number of steps per period to reach a very good accuracy, typically better than $10^{-4}$.

Figure 3

Referee is $k$ the same integer as $p$? If yes, better use $p$.

Authors We are a bit confused by the comment. Probably the Referee has mistaken $\kappa$ (defined in Eq. (11)) for $k$ ?

References

Referee non-abelian anyons have recently been observed on Honeywell's trapped ion quantum computer: https://www.nature.com/articles/s41586-023-06934-4

Authors We thank the Referee for this interesting paper which we have cited in the introduction.

Referee the authors might be interested in a recent preprint, where we discuss various aspects of state preparation under strong Floquet drives, and in particular how to speed up adiabatic ramps: https://arxiv.org/abs/2310.02728

Authors We thank the Referee for pointing out this interesting preprint. We have cited it in the final discussion.

Typos

Referee

  • abstract: remarquable $\longrightarrow$ remarkable

  • after Eq(1) [+ other instances]: "significantly smaller that the smallest": that $\longrightarrow$ than

  • before Eq(2): Krylov's $\longrightarrow$ Krylov

  • Eq(2): summation index m missing in sum subscript

  • Sec 2.2, first paragraph, "It can be view as": view $\longrightarrow$ viewed

  • Sec 3.1, penultimate paragraph, "p fixed": p $\longrightarrow$ p

  • Sec 3.2, first paragraph, "is suddenly switch on", switch $\longrightarrow$ switched

  • Sec 3.4.2, last paragraph, "As shown in Figs. 7": Figs $\longrightarrow$ Fig

  • Sec 4.3: "We argue that our goal cannot be realized by branching the drive suddenly": branching $\longrightarrow$ quenching

  • Sec 4.3: "the minimum ramp time to reach a good fidelity should qualitatively scales like N": scales $\longrightarrow$ scale

Authors We thank the Referee for pointing out the typos which have all been correcte]d.

Added references

Note that Refs. [19], [23] and [74] have been added.

---

## Round 3 · Referee Report · Marin Bukov (Referee 1) · 2024-5-4

Report
I thus recommend publication without further delay.
Recommendation
Publish (easily meets expectations and criteria for this Journal; among top 50%)

Author: Matthieu Mambrini on 2024-05-30 [id 4524]
(in reply to Report 2 on 2024-05-25)Responses to the 2nd referee:
We thank the Referee for his/her insightful reading of the paper. We have made small revisions and added the requested references following his comments. Below we answer point by point and describe the changes made.
Referee
The adiabatic evolution starts from a simple low-entangled SU(2)-symmetric initial state, which is then evolve adiabatically into a CSL. This process will cross a quantum phase transition, where the gap closes. How can adiabaticity be ensured? How does the required ramp time scale with system size? If possible, it might be helpful to perform a finite size scaling of the fidelity.
Authors
We would like to emphasize that it is essential that the system is kept finite and we believe the thermodynamic limit cannot be taken right away for two reasons: first, heating will occur when the many-body spectrum (scaling with system size as N/$\omega$) will "touch" the boundary of the Floquet-Brillouin quasi-energy zone of extension $\omega$. This simple argument gives a minimum frequency which diverges as $\sqrt{N}$ for increasing system size; secondly, as mentioned by the referee, another issue is the vanishing of the finite size gap in the thermodynamic limit that would probably lead to a diverging ramp time. However, we believe our set-up is still relevant for experiments with a finite number of qubits. This comment has been added as a small paragraph in the "Final remarks" subsection.
Referee
In Rudner et al [58], it is argued that there exists a non-equilibrium phase with chiral edge states with Chern numbers zero in the limit of a slow drive. This was then generalized to a Kitaev type model in Po et al. Phys. Rev. B 96, 245116 (2017). Starting from your model, would it be possible to find a similar phase in an SU(2) symmetric model?
Authors
We thank the Referee for mentioning the reference that we have added. Indeed, moving away from the high-frequency limit is a very interesting problem. In fact, this is a new project we are investigating at the moment, looking for "anomalous CSL". We have mentioned this possibility in the "Final remarks" subsection.
Added references
Note that Ref. [76] has been added.

---

## Round 3 · Referee Report · Anonymous (Referee 2) · 2024-5-25

Strengths
Weaknesses
- The numerics is shown only for one system size (4x4 Torus).
Report
The manuscript is overall well written, fairly self contained, and all results are presented clearly. The numerical results appear to valid and are of high quality.
While reading the manuscript, a few comments/questions came to my mind that the authors could address:
• The adiabatic evolution starts from a simple low-entangled SU(2)-symmetric initial state, which is then evolve adiabatically into a CSL. This process will cross a quantum phase transition, where the gap closes. How can adiabaticity be ensured? How does the required ramp time scale with system size? If possible, it might be helpful to perform a finite size scaling of the fidelity.
• In Rudner et al [58], it is argued that there exists a non-equilibrium phase with chiral edge states with Chern numbers zero in the limit of a slow drive. This was then generalized to a Kitaev type model in Po et al. Phys. Rev. B 96, 245116 (2017). Starting from your model, would it be possible to find a similar phase in an SU(2) symmetric model?
In summary, this work contains novel results and presents an innovative algorithm. However, the author should address the comments above before I can recommend publication in SciPost Physics Quantum.
Recommendation
Publish (meets expectations and criteria for this Journal)

---

## Round 3 · Author Response

We would like to resubmit a new version of the draft and enclose below the response to the 1st referee. We believe that the changes made to comply with the comments of the 1st referee have improved the paper significantly. We are awaiting for the report of the second Referee who can then benefit from the improved version.
Sincerely yours,
M. Mambrini and D. Poilblanc

---

## Round 3 · List of Changes

We thank Prof. Marin Bukov very much for is insightful reading of the paper. We have made some revisions and added some references following his comments. We believe it has improved the quality and clarity of the paper. Below we answer point by point and describe the changes made.
Section 2
Referee Eq(3); it would be helpful if the authors define explicitly the permutation operator in terms of the spin degrees of freedom (I now see that you sort of have it in Eq(24) but an explicit reference will be useful).
Authors A sentence has been added to make the definition of $P_{ijkl}$ explicit. The expression in terms of the spin degrees of freedom has now been included in the main text.
Referee Sec 2.3, "Note that such a monochromatic drive may not be easy to implement on experimental cold atom platforms": I think there is no difficulty with the dephased monochromatic drive itself; the spin-spin interactions would be the real challenge, see e.g., https://www.nature.com/articles/s41567-020-0949-y
Authors We thank the Referee for mentioning the reference. We have rephrased the short paragraph in Sec. 2.3 on the experimental implementation, taking into account the Referee comment.
Referee Sec 2.4: before Eq (6), when the authors introduce stroboscopic times, there appear integers p, n, times t, t0, etc,; what's written is not wrong, but I think it can be simplified by removing the unnecessary info for the benefit of the readers.
Authors Following the comment we have simplified the formulation (removing $t_0$, etc...).
Referee after Eq(7), "It is interesting that the next order of the expansion is $O(1/\omega^3)$ so that we expect HF0 to capture accurately the stroboscopic motion.": this is not obvious to me; can the authors argue why there are no $1/\omega^2$ terms here?
Authors Eq. (42) of Bukov et al. suggests that, in the absence of a time independent part $H_0=0$, the expansion involves only nested commutators containing the same number of $H_1$ and $H_{-1}$ Fourier components of the Hamiltonian, i.e. odd powers of $1/\omega$. We have added a note on this.
Referee If I'm not mistaken, you might be able to compute the Floquet Hamiltonian to Eq(6) alone exactly, as follows:
-
Let $L_z$ be the generator of continuous rotations about the out-of-lattice-plane axis, i.e., the operator that rotates $H_x$ into $H_y$: $\exp(-i\pi/2L_z) H_x \exp(+i\pi/2 L_z )= H_y$; I think, $\exp(-i\pi / 2 L_z)$ is related to the permutation operator $P_{i,j,k,l}$ but we need the infinitesimal rotation generator. Not sure how crazily non-local (in terms of the spin operators) $L_z$ is, though.
-
consider now a rotation about the angle $\alpha = \omega t$: in a co-rotating frame defined by the rotation $V(t)=\exp(-i \omega t L_z)$, the circular drive will freeze and point, let's say, along $H_x$.
-
taking into account the Galilean term due to the frame transformation, in the co-rotating frame the Hamiltonian will read as $H'= \omega L_z+H_x$ (no liability regarding the correctness of signs).
This business is very similar to the two-level system in a circular drive, just the operators involved are quite a bit more complicated. To relate $H'$ to HF, you may need to do extra work, especially if the spectrum of $L_z$ is not commensurate, see Sec 2.3 of Ref [39]. In the language of $H'$, to make the chiral structure manifest, you'd have to consider the "symmetric gauge", see discussion in Sec 3.3 of Ref [39].
Authors We thank the Referee for his proposal. However, we believe that, away from the high-frequency limit, the Floquet Hamiltonian becomes more and more non local (including all sorts of chiral terms on larger and larger closed loops). It would be very surprising that tractable expressions could be obtained generically. In any case, we postpone this calculation to future studies.
Section 3
Referee Sec 3.2.1: "Since the effective Floquet dynamics is expected to be very accurate, one can use it to investigate the physics in the $t\rightarrow \infty$ limit": this is dangerous b/c the approximate effective Hamiltonian computed from the Magnus expansion does not capture heating effects. The system is expected to heat up to an infinite temperature state in the $t\rightarrow \infty$ limit for any finite $\kappa>0$, see e.g., Ref [33]. However, what's physically relevant in practice is the prethermal plateau regime which occurs at finite time (and whose lifetime can be controlled by increasing the drive frequency).
Authors We agree with the referee and have reformulated the text. We have added a comment on the role of the finite system size.
Appendix A
Referee "PEPS offer an extremely efficient variational scheme to address local Hamiltonians": it's not clear which features/properties are being addressed
Authors Beside providing a very good variational energy, PEPS can encode the topological nature of the state and the physics of the edge mode. Note and Refs. [75,76] added.
Referee it's worthwhile to mention in one sentence the drawbacks of PEPS as well, or the regime of applicability
Authors PEPS fail to describe the rapid increase of entanglement entropy (e.g. in the case of a quench, see Ref. [77]) but should still be relevant in the case of an adiabatic ramp.
Appendix C
Referee "The unitary operator $\exp(-iK(t))$ corresponds physically to a change of basis": time-dependent unitaries correspond rather to a change of reference frame; when evaluated at some fixed $t_0$, one can think of them as giving kicks to the state (i.e., changing the basis). In this sense, Floquet's theorem is a statement about the existence of a reference frame, where the dynamics is governed by a static Hamiltonian $H_F$ at all times (not only stroboscopically).
Authors We have changed "change of basis" into "change of reference frame (via a change of basis)".
Figure 2
Referee a) I find it a bit confusing that the micromotion in inset
(c) seems to oscillate around the black-squares curve; however the two
curves are w.r.t. different x-axes (black squares go from times
$\sim$<!-- -->25 to 40, while grey micromotion curve goes from
times 30.1 to 30.5). It might be better to show inset (c) as a separate
panel of the figure to avoid this confusion.
Authors We have modified the figure to avoid the confusion mentioned by the Referee.
Referee b) regarding errors caused by Trotterization: I think it should be possible to evolve the state using a Runge-Kutta solver on a $4\times4$ patch.
Authors We have no expertise with such a technique but we have full control on the error introduced by the Trotterization scheme. By varying the Trotter step we can estimate the minimum number of steps per period to reach a very good accuracy, typically better than $10^{-4}$.
Figure 3
Referee is $k$ the same integer as $p$? If yes, better use $p$.
Authors We are a bit confused by the comment. Probably the Referee has mistaken $\kappa$ (defined in Eq. (11)) for $k$ ?
References
Referee non-abelian anyons have recently been observed on Honeywell's trapped ion quantum computer: https://www.nature.com/articles/s41586-023-06934-4
Authors We thank the Referee for this interesting paper which we have cited in the introduction.
Referee the authors might be interested in a recent preprint, where we discuss various aspects of state preparation under strong Floquet drives, and in particular how to speed up adiabatic ramps: https://arxiv.org/abs/2310.02728
Authors We thank the Referee for pointing out this interesting preprint. We have cited it in the final discussion.
Typos
Referee
-
abstract: remarquable $\longrightarrow$ remarkable
-
after Eq(1) [+ other instances]: "significantly smaller that the smallest": that $\longrightarrow$ than
-
before Eq(2): Krylov's $\longrightarrow$ Krylov
-
Eq(2): summation index m missing in sum subscript
-
Sec 2.2, first paragraph, "It can be view as": view $\longrightarrow$ viewed
-
Sec 3.1, penultimate paragraph, "p fixed": p $\longrightarrow$ p
-
Sec 3.2, first paragraph, "is suddenly switch on", switch $\longrightarrow$ switched
-
Sec 3.4.2, last paragraph, "As shown in Figs. 7": Figs $\longrightarrow$ Fig
-
Sec 4.3: "We argue that our goal cannot be realized by branching the drive suddenly": branching $\longrightarrow$ quenching
-
Sec 4.3: "the minimum ramp time to reach a good fidelity should qualitatively scales like N": scales $\longrightarrow$ scale
Authors We thank the Referee for pointing out the typos which have all been correcte]d.
Added references
Note that Refs. [19], [23] and [74] have been added.

---

## Round 4 · Referee Report · Anonymous (Referee 2) · 2024-5-30

Report

The authors address my minor comments and I recommend publication.

Recommendation

Publish (easily meets expectations and criteria for this Journal; among top 50%)

---

## Editorial Decision

published